# Pyfectious: An individual-level simulator to discover optimal containment policies for epidemic diseases

**Arash Mehrjou** [1,2]☯*, **Ashkan Soleymani**[3]☯, **Amin Abyaneh**[4], **Samir Bhatt**[5], **Bernhard Schölkopf**[1,2], **Stefan Bauer**[6]

**1** Max Planck Institute for Intelligent Systems, Tübingen, Germany, **2** ETH Zürich, Zürich, Switzerland, **3** Massachusetts Institute of Technology, Cambridge, Massachusetts, United States of America, **4** McGill University, Montreal, Canada, **5** Faculty of Medicine, School of Public Health, Imperial College London, London, United Kingdom, **6** KTH Stockholm, CIFAR Azrieli Global Scholar, Stockholm, Sweden

☯ These authors contributed equally to this work.
* arash@distantvantagepoint.com

**Data Availability Statement:** We have published the Pyfectious source code on github.com/amehrjou/Pyfectious. Tutorial can be found at colab.research.google.com/drive/14UYB9x0g5s7jyHO4DynqDJMXTaSM11eW. All

## Abstract

Simulating the spread of infectious diseases in human communities is critical for predicting the trajectory of an epidemic and verifying various policies to control the devastating impacts of the outbreak. Many existing simulators are based on compartment models that divide people into a few subsets and simulate the dynamics among those subsets using hypothesized differential equations. However, these models lack the requisite granularity to study the effect of intelligent policies that influence every individual in a particular way. In this work, we introduce a simulator software capable of modeling a population structure and controlling the disease's propagation at an individualistic level. In order to estimate the confidence of the conclusions drawn from the simulator, we employ a comprehensive probabilistic approach where the entire population is constructed as a hierarchical random variable. This approach makes the inferred conclusions more robust against sampling artifacts and gives confidence bounds for decisions based on the simulation results. To showcase potential applications, the simulator parameters are set based on the formal statistics of the COVID-19 pandemic, and the outcome of a wide range of control measures is investigated. Furthermore, the simulator is used as the environment of a reinforcement learning problem to find the optimal policies to control the pandemic. The obtained experimental results indicate the simulator's adaptability and capacity in making sound predictions and a successful policy derivation example based on real-world data. As an exemplary application, our results show that the proposed policy discovery method can lead to control measures that produce significantly fewer infected individuals in the population and protect the health system against saturation.

other relevant data are within the paper and its Supporting information files.

**Funding:** The author(s) received no specific funding for this work.

**Competing interests:** The authors have declared that no competing interests exist.

## Author summary

Pyfectious is an agent-based simulator with the capability to serve as an environment for reinforcement learning agents to discover novel control high-resolution agent-based policies that are hard for humans to discover. Pyfectious introduces several novelties which are unprecedented in the existing popular simulators in epidemiology. It constructs the population structure of a city without needing too detailed information by a novel probabilistic assignment method that is unparalleled to existing population synthesizers. The proposed disease propagation algorithm offers a multi-resolution functionality that allows running Pyfectious for large-population cities on normal computers. The modeling details can be easily traded-off with computational demand requiring minimal effort by the end user. The control and monitoring components are designed in an event-triggered fully flexible way by providing a rich action space from which effective policies are hoped to be discovered by advanced RL methods which are otherwise impossible for humans to find due to the immense complexity of the problem. An extensive set of experiments are included to illustrate various aspects of Pyfectious and also to briefly showcase its use as an RL environment which is hoped to help the automatic discovery of epidemic control policies upon bringing together RL scientists and epidemiologists.

## Introduction

The approaches to control an epidemic disease such as COVID-19 are divided into two main categories: **1)** pharmaceutical and **2)** non-pharmaceutical. While the first category involves medication and vaccination, the second approach, which is the main interest of the current work, concerns behavioral interventions to slow down the spread of the disease [1]. The objective of non-pharmaceutical methods is to reduce the growth rate of the infection to prevent collapsing the healthcare systems that are widely known as flattening the curve [1]. When infectious diseases cause an epidemic, strict control measures such as bans on crowded public events, travel restrictions, limiting public transportation, and minimizing physical contacts are commonly adopted by countries to control the number of infections and prevent their healthcare system overburdening [2]. These methods are collectively known as social distancing. There are other control measures that do not belong to social distancing policies but have similar effects in the sense that they help reduce the rate of the spread of the infection in the population. The major player of this group is the vaccination policy. Notice that we made a distinction between the development phase (pharmaceutical) and the application phase (non-pharmaceutical) of vaccines. The former includes efforts from biological and chemical R&D phases to clinical trials. The latter assumes the vaccine is developed and has a certain effect in reducing the probability of getting infected in every contact. Considering the vaccine policy as behavioral intervention allows investigation of situations where there is a shortage of vaccine supply or the population is not fully vaccinated along with other behavioral interventions such as social distancing.

Social distancing policies are often based on experts' common sense and previous experiences in partially similar conditions [1]. For instance, travel restrictions [3], school closure [4], and wearing protective instruments such as masks [5] were used during SARS-CoV (Severe Acute Respiratory Syndrome Corona Virus) in 2003 whose effect were evaluated through several studies.

A detailed study of the effect of various policies after an epidemic breaks out requires a precise population model. Models are developed at different levels of abstraction. Compartment

models divide the population into sub-groups and model the spread of the disease as a system of differential equations whose states are the size of each sub-group [6]. By defining more fine-grained sub-groups, the model becomes more accurate and realistic, while at the same time, it becomes computationally more demanding. In a more detailed extreme, the model states the health conditions of every individual in the population.

There are two primary benefits in modeling the population at the level of individuals: **1)** They can be utilized to verify more abstract models; that is, the collective behavior observed in compartment models must be aligned with the aggregation of the states of the fine-grained models. **2)** They allow investigating the policies influencing every individual uniquely. Therefore, more sophisticated policies can be proposed compared to compartment-level policies that equally affect all members of a particular community. For example, super-spreaders are known to have a critical role in driving a pandemic. Investigating the effect of controlling them is crucial to direct the limited control resources more effectively. This study and similar ones of fine-grained control measures are not feasible in compartment models.

We have developed Pyfectious, a lightweight python-based individual-level simulator software together with a probabilistic generative model of the structured population of an arbitrary city. This software's components are developed by having the ultimate idea that it is going to be used as a reinforcement learning environment that is fast yet detailed enough to discover non-trivial control policies for real-world pandemics in the future. We hope that the same role that the Chess rules played for AlphaZero to learn a superhuman chess player [7], Pyfectious would play for a general-purpose reinforcement learning algorithm to learn the best policy to control epidemic diseases. The software and its accompanying examples are available at github.com/amehrjou/Pyfectious.

Therefore, our work's main contributions compared to existing simulators in different aspects such as population generation, simulation, policy enforcement, and runtime details are discussed in the following paragraph. We tried to compare with a diverse set of highly-respected simulators in terms of the goal, i.e., the problem they intend to solve, and the simulation methods in different pieces of literature, e.g., epidemiology [8, 9], artificial intelligence [10], social network [11], economics [12], etc. Some of them are retrieved by running queries on the Google Scholar database to find the recent highly-cited simulations.

- **Population model**: We enumerate various aspects of generating a society by creating a set of individuals. These aspects include attributes of each individual, their roles in the society, their daily schedule, the types of interaction with each other, etc. (See Table 1 for comparison with existing software packages).

  – **Generality (referred to as detail level in the tables)**: One of the factors that distinguish the population generator component of Pyfectious from existing works such as [11, 13–16] is the capability of Pyfectious in defining different roles that determine the types of interactions within a certain location. Most existing works assign individuals to a location based on their occupation or other properties. Pyfectious goes deeper and allocates individuals to various groups with different types and intensities of interactions based on their roles. This criterion determines the detail level of the population, that is, the granularity of the simulator in modeling the attributes of individuals and their interactions in the real world. For instance, in a school, the simulator's ability in modeling students, teachers, cleaning crew, and other roles is pivotal to being faithful to the real-world dynamics of schools. Here, we exemplified some details that Pyfectious takes into account when simulating schools at this granularity level. The population of a school community is divided into roles such as students and teachers based on the individuals' attributes. For example, students of the same class belong to the same age group. The other important factor is types of interactions

**Table 1. Comparing the population model in various epidemic simulation softwares.**

| Method | Type | Detail Level | Population Model | | |
|---|---|---|---|---|---|
| | | | Probabilistic | Generative | Real-world Data Awareness |
| SNDS [11] | Social Network | Moderate | ✓ | ✓ | ✗ |
| Age-structured SEIR [15] | Compartment Level | Low | ✗ | ✗ | ✓ |
| EpiFlex [17] | Individual Level | High | ✓ | ✓ | ✓ |
| EpiSimS [8, 18] | Individual Level | High | ✓ | ✓ | ✓ |
| EpiModel [9] | Network Model | Moderate | ✗ | ✗ | ✗ |
| SAMSCE [19] | Individual Level | Moderate | ✓ | ✓ | ✓ |
| CHIME [22] | Compartment Level | Low | ✓ | ✓ | ✓ |
| TDCO [22] | Compartment Level | Low | ✗ | ✗ | ✓ |
| SCEDS [16] | Compartment Level | Low | ✗ | ✗ | ✓ |
| Diamond Princess Analysis [21] | Individual Level | Moderate | ✗ | ✗ | ✓ |
| Epidemiology Workbench [20] | Individual Level | Moderate | ✓ | ✓ | ✓ |
| How to Restart? [25] | Individual Level | Moderate | ✓ | ✗ | ✓ |
| SoEcNetwork Heterogenity [12] | Individual Level | High | ✓ | ✓ | ✓ |
| QECTTC [10] | Individual Level | High | ✓ | ✓ | ✓ |
| **Pyfectious (ours)** | **Individual Level** | **High** | ✓ | ✓ | ✓ |

among individuals of different groups. For example, interactions among the students of the same class are more frequent and effective in transmitting the disease than interactions among students of different classes. Moreover, unlike the simulators that allow individuals to take simple roles, Pyfectious is capable of modeling individuals with complicated and compound roles. For example, an individual may have a job, be a member of some friend gatherings, use public transport, eat regularly at some restaurants, be a member of a gym, and so on. The design of Pyfectious is entirely suitable to model these details.

– **Extendability**: The roles of the individuals, their corresponding daily schedule, the interactions among them, and their personal attributes are all manually configurable in Pyfectious. The format of providing these parameters as the input to Pyfectious is detailed in S1 File, where a low-level description of the simulator's functionalities is provided, and S2 File that explains an automated and descriptive simulation interface. This level of flexibility allows Pyfectious to be configured for any arbitrary city based on the real-world statistics of that city's population structure. Having maximum flexibility has been one of our initial design goals for Pyfectious to make it suitable for a wide range of applications, including discovering the optimal control measures for any target city. This level of extendability is not provided by most of the existing methods up to our knowledge, even in widely used simulators like [8, 9, 17, 18].

– **Probabilisticity**: Another fundamental design principle of Pyfectious is to make the conclusions derived from the simulation's outcomes robust against slight changes in the provided settings. Therefore, every parameter of the simulation is assumed to be a realization of a distribution that can be provided as the input configuration. Therefore, the entire structured population of the city is a large hierarchical random variable. Many cities with almost the same population structure can be sampled from this random variable to derive confidence bounds on the simulation's obtained results. Lots of existing simulators [10–12, 19, 20] lie under the umbrella of probabilistic nature.

**Table 2. Comparing the employed methods to simulate the dynamics of epidemic disease in various epidemic simulation softwares.**

| Method | Simulation Model | | | |
|---|---|---|---|---|
| | Type | Interactions Model for Disease Transmission | Detail level | Multi-resolution |
| SNDS [11] | Actor stepwise | Interaction in network | Moderate | ✗ |
| Age-structured SEIR [15] | Continuous-time | Location-based contacts | Low | ✗ |
| EpiFlex [17] | Event-based model | Location-based contacts | Moderate | ✗ |
| EpiSimS [8, 18] | Event-based model | Location-based contacts | High | ✗ |
| EpiModel [9] | Clock-based model | Interaction in network | Moderate | ✗ |
| SAMSCE [19] | Clock-based model | Network/Location contacts | Moderate | ✗ |
| CHIME [22] | Clock-based model | Contacts | Low | ✗ |
| TDCO [22] | Clock-based model | Contacts | Low | ✗ |
| SCEDS [16] | Clock-based model | Contacts | Low | ✗ |
| Diamond Princess Analysis [21] | Event-based model | Location-based contacts | Moderate | ✗ |
| Epidemiology Workbench [20] | Clock-based model | Location-based (Lattice) contacts | High | ✗ |
| How to Restart? [25] | Clock-based model | Proximity-based and Exposure-time-based contacts | Moderate | ✗ |
| SoEcNetwork Heterogenity [12] | Clock-based model | Contact matrices derived from contact network | High | ✗ |
| QECTTC [10] | Event-based model | Mobility-based contacts | High | ✗ |
| **Pyfectious (ours)** | **Clock/Event-based model** | **Graph/Location-based contacts** | **High** | ✓ |

- **Simulation**: The configuration parameters of the simulator (See Table 2) that are explained below determine the accuracy vs. computation trade-off of the simulation.

  – **Event/clock-based simulation**: The simulators of temporal events often belong to one of two categories: event-based or clock-based. In an event-based simulation [8, 10, 17, 18, 21], queues of planned events are executed by their order of occurrences. In a clock-based simulation [9, 16, 19, 20, 22], a series of periodic changes to simulator modules occur at specified time intervals. To gain more computational efficiency for the purpose of this work, we combine these two methods into a clock/event-based approach. Briefly speaking, it acts as an event-based simulation equipped with a running background clock. Even though the events in the queue are executed by the specified order, the event of interest (transmission of the disease) occurs only at the clock edges. The detailed description of this mixed proposed simulation method and its advantages come in Section 1.2.4.

  – **Multiresolution time**: The settable clock/event method that was briefly described above allows Pyfectious to be multi-resolution. The resolution is controlled by the running background clock period and is adjustable according to the available computational resources. Multi-resolution timers have not been provided in previous simulators.

  – **Interaction model**: The transmission of the disease is through an underlying graph that is determined by the structure of the population. This structure allows a detailed simulation of the individuals' mobility and interactions. The daily schedule of each individual and her exposure time to other individuals can be thoroughly modeled. Further details, such as the transmission of disease by touching a surface that an infected individual already touches, can also be modeled thanks to the flexibility of the underlying connectivity graph.

- **Policy enforcement**: To mitigate the spread of the virus and mortality, governments enforce policies to limit the potential ways of disease transmission. A special feature of Pyfectious is the possibility to construct smart policies that can act on each individual differently according to her condition. A mixture of policies with different levels of granularity can be enforced

**Table 3. Comparing simulation softwares in terms of the policy enforcement methods. Entries indicated by "-" for EpiModel [9] show that polices are not implemented yet.**

| Method | Types | Policy Enforcement | | | | |
|---|---|---|---|---|---|---|
| | | Flexibility | Probabilistic Policies | Conditional Policies | Extendability | Policy Discovery |
| SNDS [11] | -seek similarity<br>-strengthen community<br>-repeat-contact bubble | Moderate | ✓ | ✗ | ✗ | ✗ |
| Age-structured SEIR [15] | -school break and holidays<br>-school closure, stop 90% workforce | Low | ✗ | ✗ | ✗ | ✗ |
| EpiFlex [17] | -decrease infection prob. | Very Low | ✗ | ✗ | ✗ | ✗ |
| EpiSimS [8, 18] | -household quarantine<br>-therapeutic treatment<br>-school closures<br>-social distancing<br>-vaccination<br>-contact tracing | Moderate | ✗ | ✗ | ✗ | ✗ |
| EpiModel [9] | Not Implemented Yet | Moderate | - | - | ✓ | - |
| SAMSCE [19] | -lockdown<br>-physical distancing<br>-mask-wearing<br>-shielding of the population at risk | Moderate | ✗ | ✗ | ✗ | ✗ |
| CHIME [22] | -social distancing | Low | ✗ | ✗ | ✗ | ✗ |
| TDCO [22] | -quarantine individual<br>-government control | Very Low | ✗ | ✗ | ✗ | ✗ |
| SCEDS [16] | -isolation measures<br>-social distancing | Very Low | ✗ | ✗ | ✗ | ✗ |
| Diamond Princess Analysis [21] | -self-protection scenarios<br>-control scenarios | Moderate | ✗ | ✗ | ✗ | ✗ |
| Epidemiology Workbench [20] | -self-isolation<br>-social distancing<br>-testing<br>-contact tracing | Moderate | ✗ | ✗ | ✗ | ✗ |
| How to Restart? [25] | -social distancing solutions<br>-use of respiratory protective devices<br>-control of COVID-19 infectors | Moderate | ✗ | ✗ | ✗ | ✗ |
| SoEcNetwork Heterogenity [12] | social distance policies (change the structure of network) | High | ✗ | ✗ | ✗ | ✗ |
| QECTTC [10] | -lockdown<br>-contact tracing<br>-localized interventions | Moderate | ✗ | ✗ | ✗ | ✗ |
| **Pyfectious (ours)** | **Almost Every Possible Policy** | **Very High** | ✓ | ✓ | ✓ | ✓ |

during the simulation; at the same time, the statistics of the disease are fed back to the controller and updated the policy concurrently. A diverse set of built-in policies is provided with Pyfectious, and they can be a good starting point for a user to modify and investigate the result of an arbitrary policy. This feature makes Pyfectious a fast and full-fledged environment for a reinforcement learning algorithm to infer optimal policies given a specified cost function such as mortality, active infected cases, etc. To showcase this possibility, an experiment for policy discovery is provided in Section 2.2.6. The features mentioned above are summarized in Table 3.

– **Flexibility**: The design architecture of Pyfectious gives maximum flexibility for the policy design in terms of what features can be altered by a policy. Any dynamic change in the connectivity graph, testing resources, disease, and individual attributes is possible. Hence, in addition to common real-world policies such as contact tracing, social distancing, testing, vaccination, closures, quarantines, and full or partial lockdowns, more complicated and individual-specific policies can be easily studied. Intervention space even in well-establish approaches is limited to a predefined set. For instance, [8, 18] confined their possible interventions to therapeutic treatment, school closures, social distancing, vaccination, and contact tracing; or [20]'s set of available interventions is characterized by self-isolation, social distancing, testing, and contact tracing; while some complicated multi-stage counterfactual policy might lead to better results since the loss landscape of the problems is not well-shaped. To address this issue, [12] provided the possibility of changing the structure of the underlying network as possible intervention space, but even this way is not rich enough compared to Pyfectious. Besides, it rises the question of how to interpret good policies determined by specific weights in a factorized graph matrix in real-world applications. Pyfectious allows a much richer space of possible policies, including probabilistic and conditional control which does not suffer from interpretation ambiguities.

– **Probabilistic control**: Probabilistic control measures are allowed in Pyfectious. For example, a policy may enforce quarantining randomly chosen 50% of school's students each day.

– **Conditional policy**: Conditional policies are critical for smart and efficient control of an epidemic disease. In the terminology of control theory, this resembles feedback controllers where the action applied to the system depends on the observations from the systems' states. This feature allows modeling real-world policies such as the closure of public places when the number of active cases increases and re-opening them when it decreases. Examples of these feedback policies are discussed in Section 2.2.5.

– **Extendability**: A simple user-friendly language is developed to write policies of interest or extend the wide set of built-in policies. Each policy consists of two components: The condition that triggers the policy and the control measure, which is the action taken by the policy when the triggering condition is satisfied. Extendability of the possible policy space is vital for a simulation to be able to capture optimum policies, but to our knowledge, this characteristic is not properly covered in previous works.

– **Policy discovery**: By defining a cost function, e.g., the peak of the confirmed infected cases, a wide range of policies can be tested in parallel at each round and use their outcome as a learning signal for the RL agent to move in the space of feasible policies towards those with more desirable results. As an example, in Section 2.2.6, the optimal policy is inferred using Bayesian optimization when the peak of the curve of confirmed cases is taken as the cost function. Similarly, control theory and reinforcement learning algorithms can be applied for optimal policy discovery.

• **Implementation overview**: Thanks to the aforementioned novelties in the implementation at multiple levels from algorithms to software architecture, Pyfectious achieves superior scalability in the size of the simulated population and also the simulation's duration compared with other simulators (see Table 4). Comprehensive documentation and code snippets of the use cases are provided as Jupyter notebooks to facilitate a quick start in running experiments with an arbitrary setting for epidemic researchers or policymakers.

**Table 4. A comparison of the simulators considering different technical aspects.**

| Method | Technical Details | | |
| --- | --- | --- | --- |
| | Sizewise Scalable | Timewise Scalable | Language |
| SNDS [11] | ✗ (500–4000) | ✗ | R |
| Age-structured SEIR [15] | ✓ | ✓ | R |
| EpiFlex [17] | ✓ | ✓ | Windows Software written in C++ |
| EpiSimS [8, 18] | ✓ | ✓ | C++ |
| EpiModel [9] | - | - | R |
| SAMSCE [19] | ✓ | ✓ | C++ |
| CHIME [22] | ✓ | ✗ | ? |
| TDCO [22] | ✓ | ✓ | Python |
| SCEDS [16] | ✓ | ✓ | Fortran |
| Diamond Princess Analysis [21] | ✗ | ✗ | ? |
| Epidemiology Workbench [20] | ✓ | ✓ | Python |
| How to Restart? [25] | ✗ | ✗ | R |
| SoEcNetwork Heterogenity [12] | ✓ | ✓ | R |
| QECTTC [10] | ✓ | ✓ | Python |
| **Pyfectious (ours)** | ✓ | ✓ | **Python** |

This paper aims to describe the novelties of the proposed lightweight and scalable simulator software called Pyfectious. The software is designed with the ultimate goal to become a fast environment for a reinforcement learning agent to discover detailed and effective individual-level policies to control the spread of the disease in a structured population. An extensive set of experiments shows the application of Pyfectious in simulating the disease's dynamics in the population, testing the effectiveness of expert-designed policies, and automatic discovery of effective policies. The components of Pyfectious can be used independently for purposes beyond the provided examples. For instance, its population generator could be an alternative to the existing population synthesizers such as [13] for those who are more comfortable with Python than R. Moreover, as one of the goals of the design of Pyfectious have been its use as an RL environment for developing policy discovery algorithms, we believe its pythonic implementation facilitates its adoption in the machine learning and reinforcement learning communities which are dominated by pythonic frameworks such as Tensorflow [23] and PyTorch [24].

We have separated two processes in the developed software package. The first process, called Population Model, concerns the constant part of the simulation process. It creates individuals with specified features and divides them into subsets to model the population structure of the city of interest. The other process, called Propagation Model, takes the properties of the disease and the dynamic interactions among the generated individuals to evolve the states of the population model in time.

## 1 Design and implementation

The design principles of Pyfectious and our proposed methods to achieve each of them are described in detail in this section.

### 1.1 Population and the propagation models

The design principle behind the population generator of Pyfectious is the generality at all levels that makes it adaptable to an excessively wide range of populations, even when the exact

parameters of the city are not known. This is in contrast with the existing population synthesizers such as [13] that the method is designed for a certain case or needs an almost complete description of the population parameters. Even though it might be possible to change the hyperparameters of a simulated city manually, it may not be straightforward to transfer the simulator to populations from which we only have partial knowledge or are uncertain about some of their features. We develop a probabilistic model for every feature of the population, rendering it a fully probabilistic generative model.

To explain the logic behind the developed generative model for the population, we first introduce some terms and their role in the software. The full description of each term is discussed in Section 1.2.1.

1. `Population Generator`: This object is instantiated from a class called `Population Generator` and will be a primary container that stores the information required to generate a population, e.g., people, families, and communities. The `Population Generator` is a wrapper around the items inscribed below.

2. `Person`: The most fundamental object acting as the building block of the population by representing an individual is an object instantiated from the class called `Person`. The object also contains the attributes related to an individual, e.g., age, gender, and health condition.

3. `Family`: Every family is an instance of the class called `Family` and is a group of multiple individuals (modeled by `Person` objects) that live together in the same house. Each family's general composition is described by a family pattern that is itself an instance of the class `Family Pattern`. A family pattern object comprises necessary attributes to generate a `Family`, such as the number of family members and their gender, age, and health condition. Similar to any other attribute in Pyfectious, these attributes are also provided as probability distributions rather than single values. Therefore, every family pattern can be sampled multiple times to generate a set of families with an almost similar pattern but distinct values for their members' attributes.

4. `Community`: An instance of the `Community` class that describes a social unit consisting of individuals with a commonality, particularly in time and location. It is defined by a community type object and inductively by its smaller social units called subcommunities. A community type object is an instance of a class called `Community Type` that describes the attributes and subcommunities and the community's connectivity graph. A subcommunity is an instance from the `SubCommunity` class and comprises people with the same role in the community (for example, the subcommunity of teachers in a school community). A connectivity graph is an instance from the class named `Connectivity Graph` that represents the possible interactions among the individuals in the community.

Once the population is generated, the individuals' dynamic interactions and the features of the disease yield a model of the propagation of infection in the population. The essential factors in the propagation model are listed below and will be explained in more detail in the subsequent sections.

1. `Disease Properties`: Maintains the information related to the key characteristics of the disease that affects its propagation in a structured population. These properties include infection rate, immunity rate, mortality rate, incubation period, and disease period. These quantities are typically sampled from predetermined probability distributions, leading to a stochastic representation of the disease.

2. `Simulator`: The simulator employs the propagation features of the disease and the population model to evolve the simulation in time. Moreover, similar to every control task, the containment of an infectious disease demands both measurement and control. To emulate real-world processes, we develop two classes named `Command` and `Observer`. The former class instances are objects that mimic a single control decision (for example, shutting down schools if the number of infected students surpasses a threshold). The latter class instances mimic the measurement and monitoring processes, such as testing to find dormant infected cases. Both command and observer objects need a starting time. As a general solution, we built a class named `Condition`. Each instance of this class gets activated when a defined condition in the population is met. The binary output of this object can be fed into any control or monitoring option that is supposed to be triggered when this condition is satisfied.

3. `Time`: The chronological flow of the simulation is mainly based on a queue of events, and the propagation of the disease occurs at the edges of a background timer (specific time points in the time axis) whose frequency trades off accuracy versus computational demand.

    a. `Event`: The time evolution of the system is implemented in an event-driven paradigm. A sequence of events determines the daily interactions among individuals. The connectivity of the population is updated when an event occurs. Three types of events are defined in Pyfectious:

        i. `Plan-Day Event` occurs at the beginning of each day and sets the schedule of that day for every individual belonging to the population.

        ii. `Transition Event` occurs at times indicated by a plan-day event, and it changes the location of an individual.

        iii. `Virus Spread Event` occurs when the virus propagates to an individual.

        iv. `Infection Event` is queued once an individual is infected and is triggered once when the infection of an individual comes to an end.

        v. `Incubation Event` is queued upon the beginning of the incubation period and is triggered when the incubation period is over, indicating a transition from the incubation period to the illness period during which the patient is infectious.

    b. `Infection`: This is an object associated with every infected individual and keeps track of the disease-related information. The infection object must not be confused with the infection event. The former is a container that is created for every individual when she gets infected and contains all infection-related information during the course of the disease, including the outcome, which can be a recovery or death. On the other hand, an infection event is an event object that occurs when an individual's infection ends, and she can no longer infect other individuals.

In the subsequent sections, the details of the novelties in the architecture of Pyfectious are presented. A detailed description of the probabilistic algorithm that generates the population and the event-based algorithm that evolves the simulation are discussed. To follow the details, knowing the definitions mentioned above of the objects are assumed.

## 1.2 Software architecture of Pyfectious

In this section, the architecture and technical details of the simulator software Pyfectious are presented. It was mentioned earlier in the Introduction that Pyfectious consists of a population

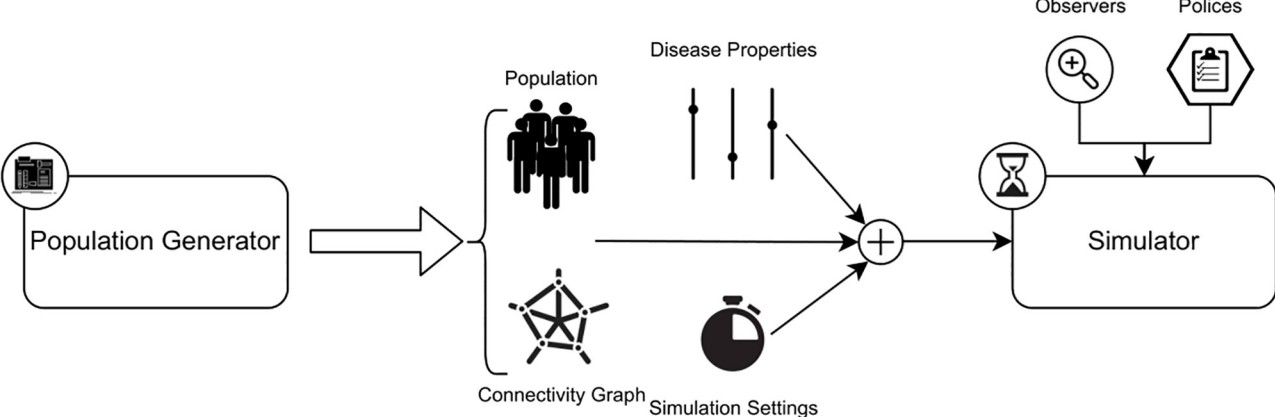

**Fig 1. The pipeline of Pyfectious.** The population generator creates the individuals and assigns them their roles to establish the connectivity graph. The connectivity graph, the disease properties, the clock, and the simulation settings are then fed to the simulator to create the time evolution of the disease. Furthermore, the observers and the policies are provided to the simulator in order to log data and make specific alternations to the simulation to emulate real-world epidemic control measures.

model and a propagation model. In a more detailed architectural description, we can consider three components that can be thought of independently: **1)** population generation, **2)** disease propagation, and **3)** time management. The first component generates the structure of a city containing a certain number of individuals that form different communities such as households, schools, shops, etc. The second component determines the properties of the disease and the way it propagates through the population. The third component ties the previous two components together to produce the evolution of a specified disease in a population with a specified structure. The overall pipeline of the software is illustrated in Fig 1. Each process is explained in detail in the following subsections.

**1.2.1 Population generation.** Generating the population consists of creating a structured set of individuals together with a connectivity graph that models the interactions among them. The connectivity graph essentially shows the possible paths where the virus can transmit among individuals. Whether the graph is directed or undirected depends on the properties of the virus for which the simulation is performed.

For the type of viruses that are still infectious after being on a surface for a while, the connectivity graph must be directed. The infected person who touches the surface at the time $t_0$ can infect the person who touches the surface at a time $t > t_0$, but the other direction of virus transmission is clearly blocked due to the time direction. The more common way of virus transmission, especially for respiratory diseases, is via close interactions while being in the same location. In this case, the connecting edges are symmetric (bidirectional). These two ways of virus transmission are illustrated in Fig 2. Regardless of the edge direction, the

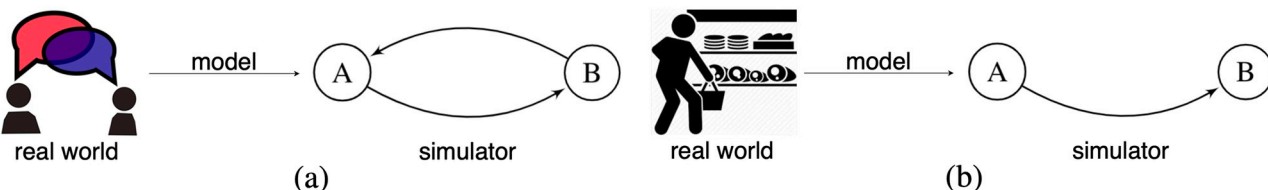

**Fig 2.** a) Two people talking to each other (bidirectional connectivity). Each person can infect the other one b) A person touching an item that was touched before by another person is modeled as single-directional connectivity. Only the first person can infect the second person.

connectivity of the population graph determines the structure of the city. The creation of such a structure is presented in Section 1.2.1. The connectivity details, such as the direction and the strength of edges are then presented in Section 1.2.2.

**Person attributes**. A human individual is realized as an instance of a class named `Person`. A person object represents the most fundamental unit of the simulation and has the succeeding attributes: age, gender, and health condition; These attributes are used to determine the role of the individual in the family and society, her hobbies, and her daily plan. Here, we list the description of these attributes of an individual. Notice that the fundamental logic behind Pyfectious is to create a significant hierarchical probability distribution for the city so that the whole city can be seen as a random variable. Hence, every property is indeed a realization of a probability distribution.

- age ($a \sim A, a \geq 0$.): A real positive number realized from a probability distribution $A$.

- gender ($g \sim G$): A binary value realized from a binomial distribution $G$.

- health condition ($h \sim H, h \in [0, 1]$): A real number realized from a specified distribution $H$ supported on $[0, 1]$ that represents the health condition of the person as an aggregated function of medical variables such as body mass index (BMI), diabetes, background heart disease, blood pressure, etc. Greater $h$ signals a better aggregated health condition.

**Family pattern**. A class named `Family Pattern` represents the pattern of the families who live in society. The pattern of a family consists of three types of information. Firstly, the structural information, i.e., the existence of each of the parents and the number of children. Secondly, the distributions from which the attributes (see Section 1.2.1) of the family members are sampled. Lastly, the distribution from which the living location of the household within the city is sampled. Let $(S, L, \{\varphi_1, \varphi_2, \ldots, \varphi_n\})$ represent a family pattern where $S$ is the structural information, $L$ is the distribution of the family location, and $\{\varphi_1, \varphi_2, \ldots, \varphi_n\}$ is the set distributions over the attributes of $n$ members of the family. Each $\varphi_i$ is itself a set of distributions of each attribute of a family member, i.e., $\varphi_i = [A_i, G_i, H_i]$. Let $p_i$ refer to the $i$-th member of the family. Its attributes are then sampled from $\varphi_i$, independent of the other members.

$$(p_i \sim \varphi_i) \equiv (p_i := [a_i \sim A_i, g_i \sim G_i, h_i \sim H_i]). \tag{1}$$

**Generating the population**. To generate the population of the city, two pieces of information are requested from the user: **1)** total size of the population **2)** a set of family patterns with the probability of their occurrence. Recall that every level of the city hierarchy is probabilistic in Pyfectious. Therefore, each family pattern can also be regarded as a random variable from which the family instances are realized. The instantiation process continues until the total number of people in the society exceeds the population size provided by the user.

Let $\mathcal{P} = \{\Phi_1, \Phi_2, \ldots, \Phi_k\}$ and $\{\pi_1, \pi_2, \ldots, \pi_k\}$ be the set of family patterns and their probabilities, respectively. Each family in society is a realization of one of these patterns. To instantiate a family, first, a family pattern is chosen with its corresponding probability, then a family instance is realized from the chosen pattern. Hence, an instantiated family pattern $\phi$ in the society follows a mixture of distributions of each family pattern in $\mathcal{P}$, that is $\phi \sim \sum_{i=1}^{k} \pi_i \Phi_i$ where $\{\pi_i\}_{i=1}^{k}$ is a simplex and $\sum_{i=1}^{k} \pi_i = 1$. This indicates that each family pattern $\phi$ follows $\Phi_1$ with probability $\pi_1$, $\Phi_2$ with probability $\pi_2$, $\ldots$, and $\Phi_k$ with probability $\pi_k$. Until the sum of the people in the families exceeds the provided population of the city, new families are kept adding to the city. Each time, one of the $\{\Phi_1, \Phi_2, \ldots, \Phi_k\}$ patterns with their corresponding probabilities $\{\pi_1, \pi_2, \ldots, \pi_k\}$ is chosen, and its members are generated from the corresponding pattern. Unlike [10], this strategy prioritizes creating families over individuals. The advantages

of this approach compared to those that create individuals first and then assign them to families are discussed below.

**Family first vs person first**. The structured population of existing real-world society is the end product of passing many generations over the years. Hence, the most accurate approach to model the current state of society is to emulate the entire time evolution from the very beginning of the formation of the city until the current time. The emulation is possible only if the emulator is given an accurate account of all major events that have occurred over hundreds or thousands of years with a significant impact on the population structure. This information is obviously unavailable at the present time. Hence, to emulate the current structured population of a society, a membership problem needs to be solved at multiple levels. Let's focus on the structure of the population at the level of families and ignore other structures such as workplaces, schools, etc. Recall that the only information we get from the user is the population size and the family patterns. One approach would be generating as many individuals as the requested population size, with attributes defined by the set of family patterns. Once this pool of individuals is created, an immensely heavy importance sampling process needs to be solved to bind individuals that are likely to form a family under the mixture distribution of family patterns. To tackle computational intractability, we propose an alternative method that puts families first and creates individuals that already match a family. This method releases us from the computationally heavy importance sampling process at the cost of having less strict control over the population size. However, the resultant population does not exceed the provided population by the user more than the size of the largest family defined in the family patterns. Clearly, one family more or less in an entire society does not alter the results of the simulations for the problem for which this software is developed.

**Algorithm 1**: Generating the population

```
Data: Population size M, Family patterns {φ₁, φ₂, ..., φₙ}, Pattern
Probability Multinomial(π₁, π₂, ..., πₖ)
Result: A structured society consisting of almost M individuals
instantiated from the Person class and distributed to families.
1 begin
2    Society = []
3    while i < M do
4       j ← generate a sample from Multinomial(π₁, π₂, ..., πₖ)
5       φ ← generate the set of family members from the pattern φⱼ
6       Society.append(φ)
7       i ← i + |φ|
8    end
9 end
10 return Society
```

**Community**. The class `Community` consists of a set of `Person` objects with a shared interest that makes them interact closely. The class `Family` is the simplest class inherited from `Community`. The concept of community in this software covers a wide range of real-world communities. It can be as small as two or three friends talking to each other, or it can refer to larger entities, such as everyone who walks in the streets of a city. Each `Community` object consists of multiple `SubCommunity` objects each of which contains a subset of the members of the enclosing `Community` that belong to that subcommunity.

As an example, every school is an instance of the `Community` class. The concept of the school contains two main roles, students and teachers, each of which is a `SubCommunity` object of the school community. An overview of this hierarchical structure is depicted in Fig 3 and its benefits are explained below.

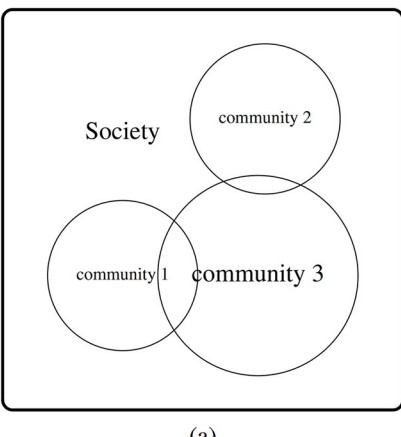 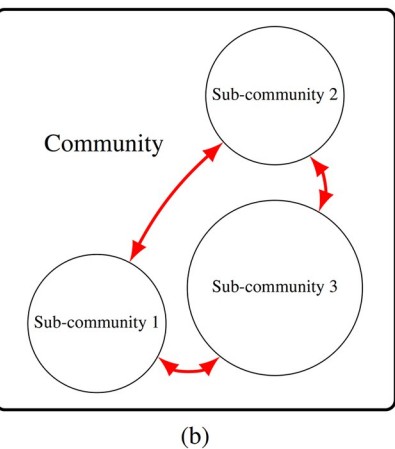 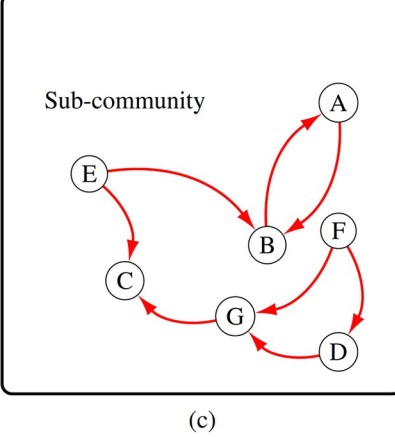

(a)                                        (b)                                        (c)

**Fig 3.** (a) In the first phase, all individuals of the society are created based on the population size and the set of provided family patterns, as explained in Section 1.2.1. (b) The abstract hierarchical structure of the communities and subcommunities of the society is created. Observe that communities may be intersecting as an individual may be a member of various communities such as family, school, restaurant, etc. Within a community, there are two types of interactions shown as red arrows. One type of interaction is inter-subcommunity, such as the interactions among teachers and students in a school. Note that, inter-subcommunity edges are indeed directional and here for ease of depiction are shown in this way. (c) Another type of interaction, shown with red arrows, is within a subcommunity. For example, the way students interact with each other in a school.

**The advantages of a hierarchical structure**. The members of a community are assigned to subcommunities based on their shared roles. In the following, we provide a list of reasons that explain the logic behind such division:

1. The members of every subcommunity in a given community have special attributes. Hence, the individuals of a society who are instantiated from a class `Person` need to pass through different filters to be assigned to each subcommunity.

2. Each subcommunity has a special pattern of internal interactions. As a result, separating them allows us to capture their influence on the spread of the infection more realistically. For example, in a school, teachers have different kinds of internal interactions compared to internal interactions among students.

3. Each subcommunity may have its own daily time schedule, even though all belong to the same community. For instance, in a restaurant that is an instance of a community, the time that a cashier spends in the restaurant is different (much longer) than the time that a customer spends there. Hence, the risk of getting infected in the restaurant is much higher for the cashier compared to the customer.

**Community assignment**. Once the needed `Community` and `SubCommunity` classes for a target society are constructed, the individuals who are instantiated from the `Person` class in Section 1.2.1 take their role by being assigned to the instances of `SubCommunities`. The challenge is that the assignment process is not trivial in the sense that we cannot fill the subcommunities from top to bottom by an ordered list of individuals. Each subcommunity accepts people whose attributes belong to a certain range. For example, the subcommunity of students in a particular school accepts individuals whose `age` attribute is less than those that are acceptable to the subcommunity of teachers.

Each subcommunity has its own set of special admissible attributes. A trivial assignment process would be an exhaustive search over the entire population to find the individuals whose attributes match those of the target subcommunity. In addition to the time intractability of this approach, the individuals may race for positions in some subcommunities while other

subcommunities do not receive sufficient individuals. Hence, we developed a stochastic filtering approach inspired by importance sampling, where the importance score is determined by how fit an individual's attributes are for a specific subcommunity. Therefore, it is helpful to view each subcommunity as a probabilistic filter that passes its matched attributes with higher probability. Individuals are assigned to their roles in society by passing through a number of these stochastic filters. To decide whether an individual can be accepted to a subcommunity, the unnormalized density of the joint attributes is computed as a fitness score.

Consider the subcommunity $S$ and the individual $p$. Assume $S$ has the $L$ admissible types of attributes and their corresponding probability densities $\{f_1, f_2, \ldots, f_L\}$. Suppose the individual $p$ has the set of attributes $\{\alpha_1, \alpha_2, \ldots, \alpha_L\}$ matching with the types of attributes that are admissible to $S$. Thus, the fitness score of $p$ for the subcommunity $S$ is calculated by $\prod_{l=1}^{L} f_l(\alpha_l)$. Notice that this score is not a probability, and computing its normalizing constant is intractable. However, this is not a problem because only the relative values matter. The scores are computed for all individuals in society, and those with the highest scores are assigned to each subcommunity.

Among all attributes, the profession of an individual requires special treatment, as discussed below. Plus, we also need to consider a hospital community that accommodates a certain number of patients during their period of infection, based on a given probability distribution.

**Special case of profession assignment**. Among the attributes of an individual, the profession needs special treatment. Every member of society can be given only one profession in the simulation period, while the other attributes can change over time. Hence, once the profession is assigned to an individual, she cannot be given another profession. Assigning individuals professions (which are essentially subcommunities) is based on their fitness score for that subcommunity. Assume an individual is assigned a profession, e.g., a university professor. To make sure she will not be assigned another profession, the fitness score for that individual becomes extremely negative for all other professions (e.g., teachers, students, cashiers, bus drivers, etc.) Mathematically, the individual's fitness scores for other professions are multiplied by $(1 - 1_{\text{has profession}} \infty)$ where the variable $1_{\text{has profession}}$ becomes 1 and hence the fitness score becomes $-\infty$ once a job is assigned to the individual.

**Hospitals**. Based on the population size, one should define one or more hospitals for each simulated town. Upon definition of a hospital in the configuration files, the simulator utilizes a hospitalization rate distribution to send known infected individuals into one of these communities for a time period extracted from another distribution, called hospitalization period distribution. People inside a hospital are considered to be quarantined and remain there until the final outcome of the infection sequence occurs, that is either death or recovery.

**1.2.2 Connectivity graph.** The backbone of Pyfectious is a connectivity graph that captures the interactions among the individuals of the society. Let $G(V, E)$ be the connectivity graph with the set of nodes $V$ and the set of edges $E$. In the following, we explain how this graph is created.

1. Every individual is represented by a node of the graph (See Fig 4a).

2. Due to the tight connections among the family members, a family is modeled by a complete and directed graph (See Fig 4b).

3. A community defines the pattern of connections within and between its subcommunities. Assume the community $C$ has the set of $J$ subcommunities $\{S_1, S_2, \ldots, S_J\}$ and a $J \times J$ connectivity matrix $(c_{ij})$ with $1 \leq i \leq j \leq J$. The entry $c_{ij}$, called connectivity density, represents the probability of the existence of an edge from an individual in the subcommunity $i$ to an individual in the subcommunity $j$. Formally speaking, Let $X_{a \to b}$ be an indicator random

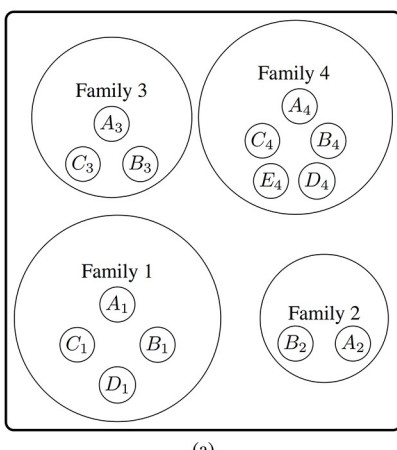 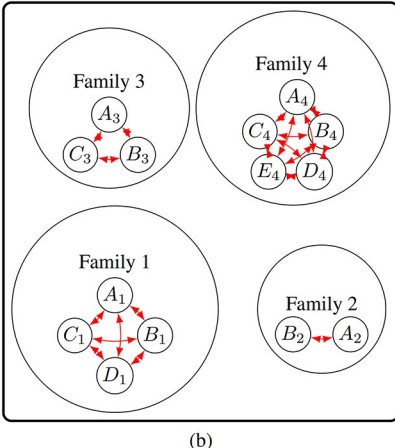 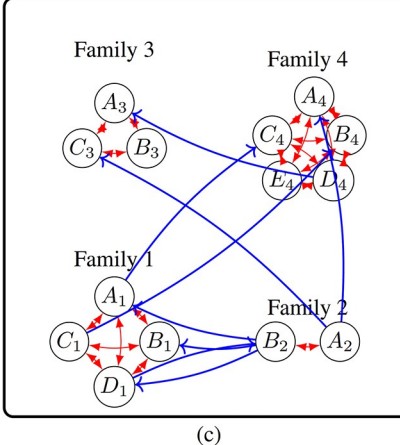

(a)                          (b)                          (c)

**Fig 4.** a) A node is added to the graph for every individual to generate the population. b) Due to the tight interactions within a family, a family's subgraph is a complete directed graph (red edges). c) The interactions across communities and subcommunities are represented by the blue edges, which are created according to Eq (2). Blue edges are directional.

variable denoting whether there is a directed edge from node $a$ to node $b$. Then $X_{a \to b}$ follows a Bernoulli distribution with parameter $c_{ij}$ if $a \in S_i$ and $b \in S_j$. The connectivity density $c_{ij}$ itself comes from a Beta distribution. That is,

$$X_{a \to b} \sim \text{Bernoulli}(c_{ij}), \quad \text{for } a \in S_i, b \in S_j, \quad \text{and } 1 \leq i \leq j \leq J \tag{2}$$

where $c_{ij} \sim \text{Beta}(\alpha_{ij}, \beta_{ij})$ with $\alpha_{ij}$ the shape and $\beta_{ij}$ the scale parameter. See Fig 4c for an overview of the created edges. The Erdős–Rényi model of generating random graphs [26] is used. Notice, that the above edge creation process does not differentiate between edges within a subcommunity and among subcommunities. However, the connections are expected to be denser within a subcommunity, that is, $c_{ij} \ll c_{ii}$ for $i \neq j$. Because $c_{ij}$ itself is a sample from $\text{Beta}(\alpha_{ij}, \beta_{ij})$, the parameters, $\alpha_{ij}$ and $\beta_{ij}$, of the corresponding beta distribution, are chosen such that the probability mass is concentrated around 1 for $i = j$ and around smaller values for $i \neq j$. The detailed pseudocode is given in Algorithm 2.

**Algorithm 2**: Creating the connectivity graph

```
Data: People {pᵢ}, families {fᵢ}, communities {Cᵢ} and their subcommu-
nities {Sᵏᵢ}.
Result: The connectivity graph G(V, E) with the set of nodes V and the
set of edges E.
1  begin
2    V ← []
3    E ← []
4    foreach person pᵢ do add a vertex vᵢ to V;
     // creating within-family interactions
5    for family fᵢ do
6      for bidirected edge e in the complete subgraph K_size(fi) do
7        add e to E
8      end
9    end
10   for community Cᵢ do
11     for sub-communities Sⁱⱼ and Sⁱₖ in Cᵢ do
12       cⱼₖ ← generate a random variable from Γⱼₖ distribution
```

```
13          p ← c_jk;
14          for person a in S_j^i and person b in S_k^i do
15            X_{a→b} ← flip a biased coin with head probability p;
16            if X_{a→b} = 1 then
               // creating inter-subcommunity interactions
17                add directed edge a → b to E
18            end
19          end
20        end
21      end
22  end
23  return G(V, E)
```

**1.2.3 Propagation of the infection.**   After the population is created in Section 1.2.1 and its structure is determined in Section 1.2.2 to model the potential interactions among every pair of individuals of the society, this section describes how Pyfectious models the propagation of a generic disease in the population.

**Disease transmission between two individuals**. The probability of the disease transmission from one individual to another depends on both the parameters of the disease and the attributes of the individuals. The following three parameters are critical in modeling disease propagation.

- Immunity ($v$): A real-valued parameter $v \in [0, 1]$ that shows how immune an individual is against being infected. For example, being infected once or being vaccinated increases this number towards the upper limit. Similar to other parameters of the model, immunity is also a random variable with an arbitrary distribution. A natural choice would be a beta distribution, i.e., $v \sim \text{Beta}(\alpha_v, \beta_v)$ whose parameters $\alpha_v, \beta_v$ needs to be determined according to the characteristics of the disease. Note that Pyfectious comes with a versatile family of distributions that can be used for any model parameter, including immunity if they are more suitable for a certain scenario.

- Infection rate ($r$): A real-valued normalized parameter $r \in [0, 1]$ that shows how easily an infection transmits in each contact. It is a unitless parameter that can be interpreted as a probability that tells how likely the transmission is once two individuals are in contact. It can also take a frequentist interpretation. For example, $r = 0.4$ means that, out of every 10 contacts, roughly 4 of them lead to a successful transmission of infection. The infection rate is a function of multiple factors such as the disease attributes (how fast a specific infection transmits) and also how well the society observes the control measures (e.g., wearing a mask, using hand sanitizer). Notice that, similar to other parameters of Pyfectious, the infection rate is a random variable whose distribution is determined by the abovementioned factors. Here we assume it has a Beta distribution, i.e., $r \sim \text{Beta}(\alpha_r, \beta_r)$ where the hyper-parameters $\alpha_r, \beta_r$ are functions of the disease attributes and the social control measures.

- Transmission potential ($\gamma$): A real-valued parameter $\gamma \in [0, 1]$ that models the possibility for the transmission of the disease between two individuals based on the type of interaction they have. Hence, this parameter is determined by the connectivity graph under the population. The individuals who meet regularly (e.g., being the members of the same family) have strong connectivity and hence a stronger potential for transmitting the infection. This parameter is also a random variable with Beta distribution whose hyperparameters are functions of the connectivity strength, i.e., $\gamma^{\text{edge}} \sim \text{Beta}(\alpha_\gamma, \beta_\gamma)$.

Given the above influential variables in disease transmission, the probability of the transmission of the infection from an infected individual (sender) to another individual (receiver)

is calculated by:

$$P(\text{sender} \xrightarrow{\text{Transmit}} \text{receiver}) = \upsilon_{\text{receiver}} \times r_{\text{sender}} \times r_{\text{receiver}} \times \gamma_{\text{edge}}. \tag{3}$$

At each interaction between infected and uninfected individuals, the above probability is calculated and kept as a threshold $p_{\text{thresh}}$. Then, a sample is generated from a uniform distribution $\zeta \sim \text{U}[0, 1]$ and a disease transmission event occurs if $\zeta \le p_{\text{thresh}}$.

**The dynamics of infection in a patient**. When an individual gets infected, several parameters are calculated based on the specified properties of the disease and the attributes of the individual. These include the period of the disease $\tau$, the probability of death $p_{\text{death}}$, the incubation period $\tau_{\text{inc}}$, hospitalization rate $p_{\text{hosp}}$, hospitalization period $\tau_{\text{hosp}}$, pre-symptomatic $\tau_{\text{psym}}$ transmission period and asymptomatic transmission probability $p_{\text{asym}}$. More specifically, the disease period is a random variable whose distribution is calculated from real-world experimental data. The same holds for all the other parameters (see the examples in Section 2.1.2 for instance). The probability distribution associated with each of the aforementioned parameters can be defined in a way to reflect other influential parameters like age and health condition. For instance, relatively high ages and poor health conditions may increase the likelihood of fatality and hospitalization.

A diseased individual contributes to the propagation of the infection until time $\tau$, based on the values of $\tau_{\text{inc}}$ and $\tau_{\text{psym}}$. An infected individual is contagious during the infection period plus a portion of $\tau_{\text{inc}}$ that is determined by $\tau_{\text{psym}}$. At the time $\tau_{\text{inc}} + \tau$, the final stage of the disease is decided as death by probability $p_{\text{death}}$ or recovery by probability $1 - p_{\text{death}}$. If the patient recovers, her attributes will be updated according to the characteristics of the infection and her attributes before getting infected. For example, temporary immunity may be gained as a result of surviving the infection once. Finally, the $p_{\text{hosp}}$ and $\tau_{\text{hosp}}$ parameters determine whether a person is kept under care in a hospital after $\tau_{\text{inc}}$ time. People staying in a hospital are released when the disease period ends while being quarantined the whole time.

**1.2.4 Time management.** A challenging issue when simulating a physical phenomenon is the immense computational resources needed to approximate the continuous evolution of the system in time. As a result, a reliable simulation becomes quickly intractable even for fairly low-dimensional systems. However, not all temporal details of the environment are relevant to the target application. When the goal is to simulate how an infectious disease propagates through a population, the only relevant events are those in which there is a potential for transmitting the disease.

Simulators often implement the evolution of time by either an event-based or clock-based method. In event-based methods, a queue of events is formed and ordered by the time-of-occurrence of them. In clock-based methods, any change in the system occurs at the pulses of a running clock. We propose a novel mixture of event/clock-driven methods to bring together the benefits of both worlds. The queue of events is formed similarly to the event-based methods, but the pulses of a background clock determine which events are effective in the outcome. The effective events are those that actually occur because glitches between two subsequent clock edges do not have any effect on the outcome. Namely, the effect of the event is reversed by another event before the next clock edge. This way, we preserve the useful details of the event-based models. By changing the frequency of the background clock, the details of the timeline can be traded off with the computational demand. Such models were used previously in [27]; however, distributing clock edges uniformly within each interval is a contribution of our method. The constituent components of the time management module of Pyfectious are explained in Sections 1.2.4 to 1.2.4 below.

**Fig 5. Yellow circles on the timeline axis indicate events.** The crossed circles represent the events that are already executed. The filled triangle represents the current simulation time. After a task is executed, the simulation time jumps to the next event in the queue. An exemplary transition is shown by moving from (a) to (b).

**Timer**. The timer object is a pointer to a specific position of the time axis during the simulation. This pointer keeps moving forward as the simulation progresses.

**Events**. An event refers to any alternation of the simulation setting, i.e., individuals' states, connectivity graph, virus spread, disease properties, etc. Every event is an instance of a class called `Event` with the following two properties: the `Activation Time` (time-of-occurrence) and the `Activation Task`. When the simulator's timer reaches the activation time, the associated task to that event (defined by the activation task) is executed. A series of events are kept in a queue and are executed in the order of their activation times (See Fig 5). To prevent the events from racing for execution, each event is given a distinct priority index as a tiebreaker in case two events happen to have the same activation time. The following events are included in the current version (V1.0) of Pyfectious. The events with higher priorities come earlier in the list: {Incubation Event, Infection Event, Plan Day Event, Transition Event, Virus Spread Event}. Each of these events is explained in the following.

**Transition event**. Each transition event is associated with a certain individual, and it is triggered when that individual changes her location from one subcommunity to another. The change of location (which is based on activities in the real world) is implemented by changing the weights of the edges connected to an individual in the connectivity graph of the corresponding community from zero to $\gamma_{edge}$ and vice versa.

**Plan-day event**. The daily dynamics of society consist of the motion of the people and their interactions based on the role they play in society. Hence, individuals' daily schedule in a city is roughly determined by their attributes and their role. In Pyfectious, the daily schedule for every individual is determined by the activation of an event at the beginning of each day. The Plan-Day event is a sequential random variable that takes the attributes and associated communities/subcommunities to an individual and generates a sample from the schedule suited to her. Looking more closely into the implementation, every daily plan is an ordered sequence of events whose start time and duration are sampled from specified probability distributions. The hyperparameters of these distributions are determined by the attributes of the individual and the subcommunities that are related to a certain event. To prevent the overlapping between the time intervals of the events, a time priority index is assigned to each event to resolve potential conflicts. For example, mandatory events, such as going to work and going to school, have a higher priority compared with optional events such as going to the restaurant.

**Incubation event**. After a disease transmission occurs, an incubation event is added to the event queue containing the end time of the incubation period. This marks the period in which the disease is still dormant in the body. When this event is activated at the end of the incubation period, the state of the respective individual will be updated to infected.

**Infection event**. After the incubation period, the respective individual's health condition is updated to infected, and an infection event is added to the event queue. This event's activation time marks the end time of the duration of the disease as a function of the individual's attributes. When this event is activated at the end of the disease period, the outcome of the disease is decided, and the individual's condition is updated to either dead or recovered.

**Virus spread event**. The running clock of the simulator is converted to a series of virus-spread events. The period of the clock works as temporal snapshots on which a virus transmission can occur. Hence, it trades off the needed computational resources with the temporal resolution of the simulator. To save computational time, the temporal resolution can be chosen long enough that assures the spread of the infection in the city does not change drastically within that period. For example, for respiratory diseases such as COVID-19, the fastest transmission way takes two individuals to get near each other. Hence, the temporal resolution can be set accordingly.

**Initializing the simulator**. To initiate the simulation, an empty queue of events is created. Then, the following two tasks are carried out to populate the queue with `Events` to be run.

1. The simulator's clock period is set, and the virus spread events are added to the queue. In Fig 6a, the clock pulses every 5 hours, and the virus spread events are placed on locations derived from a uniform distribution on consecutive 5-hour intervals starting from hour 0. For convenience, we placed the virus spread events exactly on the consecutive 5-hour intervals in Fig 6a.

2. The Plan-Day events are placed at the beginning of each day. They create the daily schedule for every individual and fill in the time progression queue with the events that make up the daily plan (See Fig 6b to 6d for illustration.)

The progress of a simple case of the simulation for two days (48 hours) is depicted in Fig 6. In Fig 6c, the plan day event is activated as the first event of the day. As a result, the transition events are added to the timeline as is shown in Fig 6d. The transition events are activated in Fig 6e and 6f that leads to changing the location of the respective individuals. A virus spread event is executed in Fig 6g and 6h and the virus transmission occurs depending on the individual's location and the connectivity graph. If a new individual gets infected as a result of a virus spread event, the infection end events are added as shown in Fig 6h. This process continues, and the added events to the timeline get executed in order until the end of the simulation time.

**Standardizing the probability of disease transmission**. We propose a multi-resolution simulator to make the simulation possible on machines with weaker computational resources. A crucial point in the multi-resolution simulator is to make sure the outcome is consistent regardless of the employed resolution (the period of the clock).

We argue that consistency is achieved if the probability of the virus transmission between two individuals is invariant with respect to the clock's period. Equivalently, it is sufficient to show that the probability of the virus not being transmitted remains invariant to the period of the clock. Hence, we equate the probability of non-transmission under clock periods $T_1$ and $T_2$ before time $t$ as

$$(1 - p_1)^{\lfloor t/T_1 \rfloor} = (1 - p_2)^{\lfloor t/T_2 \rfloor}, \tag{4}$$

where $p_1$ and $p_2$ are the probabilities of a single virus transmission. Hence, as can be seen in Fig 7, the virus transmission probability under a target clock period can be derived as a function of the transmission probability under a different clock period and the ratio of clock periods. That is

$$p_2 = 1 - (1 - p_1)^{\frac{\lfloor t/T_1 \rfloor}{\lfloor t/T_2 \rfloor}}. \tag{5}$$

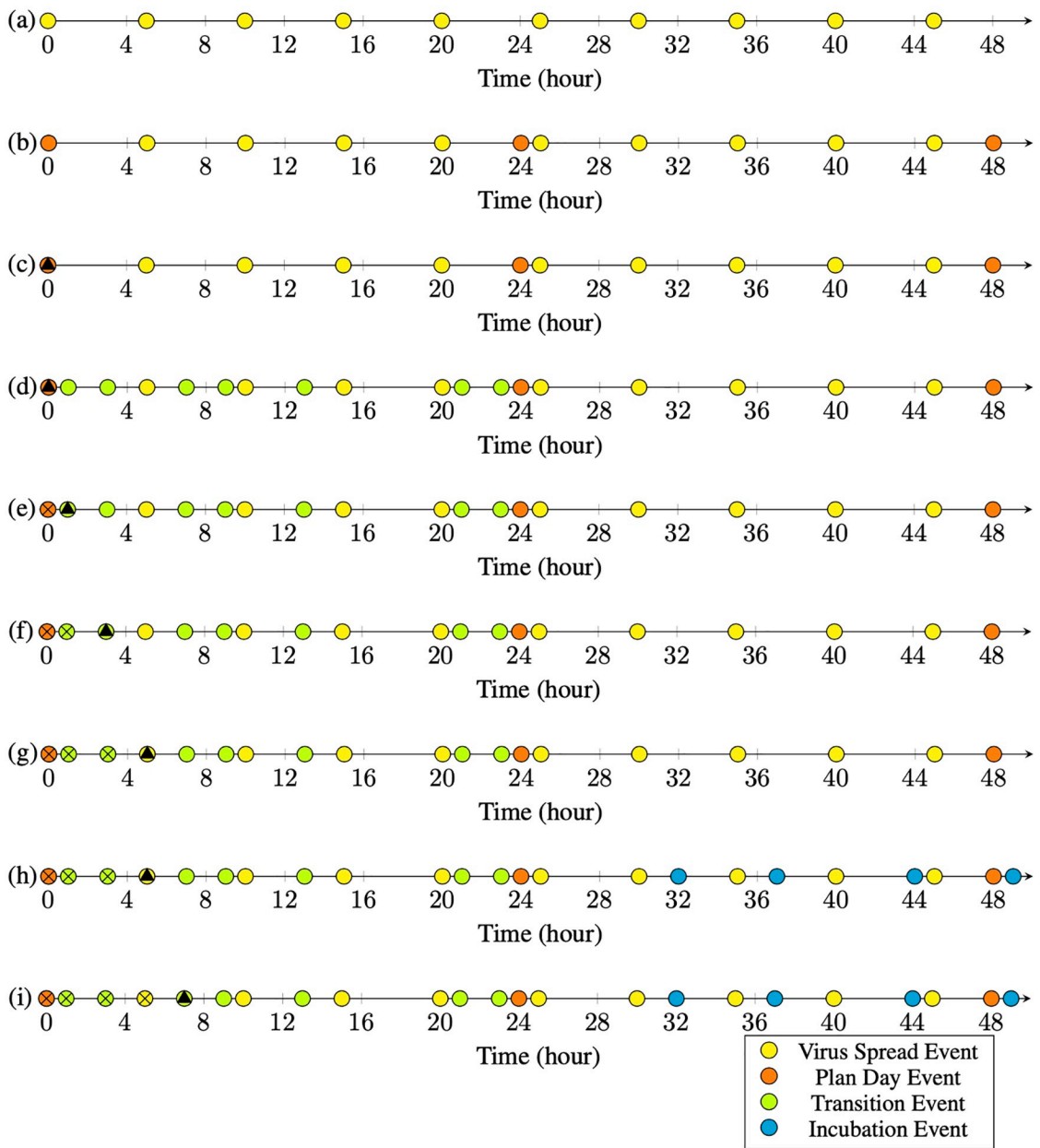

**Fig 6. In this figure, a toy example of the execution process for two days is depicted.** Events are indicated by yellow circles on the horizontal axis that represents the timeline. The current time of the simulation is shown by the filled triangle. The crossed circles represent those events that are already executed. After the execution of each event, the simulator time jumps forward to the nearest event in the queue. Each row of this figure shows one step of the simulation time. a) The simulator is initialized. The clock period is set to 5 hours based on which the virus spread events are placed. For better illustration, in this plot the virus spread events are placed exactly at the ends of the intervals; but in simulations, they are uniformly distributed within each interval. b) The planned day events are placed at the beginning of each day. These events are supposed to schedule the individuals' daily lives based on their roles in society. c) The timer is set at the start of the simulation time. d) During the execution of the planned day events, transition events are added to the event queue. These events will change the location of the individuals. e) The timer takes a step forward. f, g) The transition events are executed that change the locations of the individuals. h) The infection is being spread by interactions among individuals. New infections create new incubation events, which are added to the queue. i) The simulation moves on with the same rules.

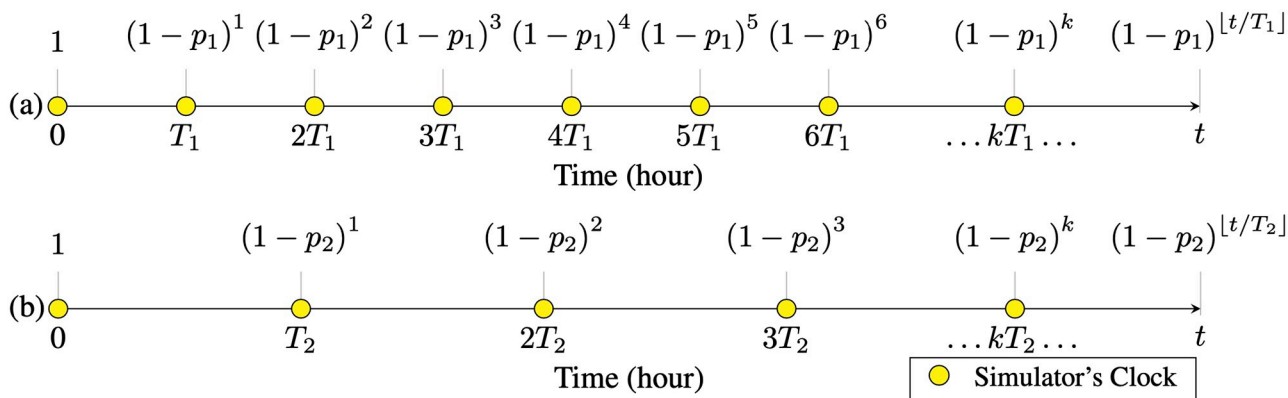

**Fig 7. Simulator's clock is translated to Virus Spread Events indicated by yellow circles on the timeline axis.** In (a) and (b) two characteristically similar simulators' timelines are displayed that only differ in their clock periods ($T_1$ for (a) and $T_2$ for (b)). If the probability of transmission in a single clock trigger is $p$, the probability of disease transmission for the whole time is a geometric distribution with parameters equal to $(p, \lfloor t/T \rfloor)$, where $T$ is the clock period. The probability of disease not being transmitted is shown above the axes for a simple interaction between two individuals. In order to have consistent results for both similar simulators with different resolutions, the mentioned probabilities should be equal to each other for the simulators. In other words, $(1 - p_1)^{\lfloor t/T_1 \rfloor} = (1-p_2)^{\lfloor t/T_2 \rfloor}$. For better illustration, in this plot the virus spread events are placed exactly at the ends of the intervals; but in simulations, they are uniformly distributed within each interval.

**1.2.5 Event management.** To emulate the real-world condition, Pyfectious is equipped with event-based measurement and control modules. These modules are triggered by the occurrence of certain events along the timeline.

**Conditions**. A condition object is instantiated from the class `Condition` and acts as a watchdog that triggers when a certain event occurs. Hence, the purpose of a condition is to notify the simulator about a predetermined event. The triggering event is a property of the condition instance. For example, one condition triggers when a specific date arrives. Another condition triggers when the number of infected people surpasses a specified threshold.

Every condition object has two standard methods. One determines whether the condition should be triggered while the other method checks if the condition has served its purposes, i.e., whether the simulator still needs the condition.

Each type of condition may have its own attributes that are implemented on demand. A list of conditions implemented in the current version (V1.0) of Pyfectious is explained below. They can be extended fairly easily by the user to support customized conditions.

- `Time Point Condition` notifies the simulator when the simulation time reaches a specified point in the timeline. Having the deadline as its parameter, the condition compares the current time with the simulation time on each pulse of the simulator's clock.

- `Time Period Condition` operates based on a given period. Starting from the beginning of the simulation, the condition notifies the simulator periodically whenever the simulator timer is divisible by the mentioned period. By increasing this divisor period, we can reduce the simulation's time resolution to reduce the computational burden at the cost of missing some short events that occur in an interval between two pulses.

- `Statistical Ratio Condition` is an example of time-independent conditions that are defined by three items: a threshold ratio, a mathematical operator, and a pair of statistics from the population. For example, let the threshold ratio be 0.2, the mathematical operator be division and the pair of statistics be the number of deaths and the number of active infected cases. Hence, this condition is triggered when more than 20% of the infected people die.

- `Statistical Family Condition` and `Statistical Role Condition` are other time-independent conditions. They follow the same logic as the `Statistical Ratio Condition` except that the given statistics that trigger the condition are taken from a specific family, role, subcommunity, or community. For example, an instance of this condition can get triggered if the number of infected students in a school surpasses a threshold. This allows interventions on specific roles within a particular community, instead of treating all members of the same role alike. For example, suppose the infection is spread in a specific school instead of shutting every school in society. In that case, Pyfectious allows investigating the outcome of quarantining the subcommunities (teachers or students) of that particular school.

**Commands**. A command object is an instance of a class named `Command`, and its function is to intervene on the other objects in a running simulation. Every command corresponds to a real-world action that changes one or more attributes in individuals, communities, subcommunities, or the edges of the connectivity graph. These actions allow the implementation of a wide range of quarantine and restriction policies. Since the command objects are designed to mimic the policies issued by the health authorities to contain the infection, they cannot change the parameters that such policies in the real world cannot alter. For example, a command can shut down a school but cannot change the inherent attributes of the disease or fixed attributes of individuals such as age and background health condition. To elaborate more, the currently implemented commands in Pyfectious are explained below.

- `Quarantine Single Person` or `Quarantine Multiple People` force an individual or a group of individuals to stay at home. The quarantine remains effective until another command lifts it. In the `Quarantine Multiple People` command, the quarantined people can be any subset of the population and do not need to belong to the same community or subcommunity. For example, if the infection is detected in the schools of a certain neighborhood of the city, a command can be issued to quarantine 50% of the students and teachers of that neighborhood's schools

- `Quarantine Infected People` puts the currently infected people in quarantine. This command operates with the idealistic assumption that there is no inaccuracy in detecting infected people.

- `Quarantine Single Family` and `Quarantine Multiple Families` put people living in the same residence in quarantine.

- `Quarantine Infected People Noisy` resembles `Quarantine Infected People` command but emulates some inaccuracy in the detection of the infected people. It also models another stochasticity in the application of the quarantine policy to better mimic real-world scenarios.

- `Restrict Certain Roles` imposes restrictions on one or more roles in the communities of the society. It has a parameter called `Restriction Ratio` that shows what ratio of the people with the target role must obey the restriction. For example, it can enforce that only 30% of the students of a school can be present on-site, and the other 70% must remain at home for a specified period.

- `Quarantine Single Community` or `Quarantine Multiple Communities` are relatively coarse-grained commands that shut down a specific community such as a particular school or restaurant.

- `Quarantine Community Type` is a coarse-grained command that shuts down all communities of the same type. For example, shutting down all the schools or restaurants of society are examples of this `Command`.

- `Change Immunity Level` is capable of increasing or decreasing the immunity level of a given set of individuals. Consequently, this command enables us to investigate the situations in which the immunity level of a specific group of individuals changes during the pandemic. In particular, as the vaccination increases the immunity level, the effect of vaccinating specific groups on changing the course of the pandemic can be helpful in designing the distribution of vaccines when the resources are limited.

- `Change Infection Rate` emulates the effect of the policies on individuals' behavior that affect how infectious a disease is. For example, in respiratory diseases transmitted via droplets while sneezing, coughing, or talking, governmental policies such as mandatory mask-wearing in certain locations result in decreasing the infection rate in those locations.

**Data logging with observers**. In reality, the virus spread information is logged with limited spatial and temporal coverage and resolution. Meaning that the information of only a subset of the population and at a sparse set of time points is logged and available to the policymakers. To emulate this real-world condition and at the same time to reduce the computational and memory usage, Pyfectious is equipped with a class named <u>Observer</u> whose instances are meant to emulate the limitations of the real-world measurements. Similar to the command objects, the observer objects are also invoked by the activation of a condition object (see Section 1.2.5). For instance, to collect the simulation data at regular intervals, a `Time Period Condition` is created to activate an observer that measures the health condition of a specific group of individuals. Every observer stores the data related to individuals, families, communities, and subcommunities in a database. Pyfectious is equipped with a comprehensive interface to get detailed reports from this database during or after the simulation is carried out for the desired duration.

## 2 Results

The concepts, novelties, and implementation details of Pyfectious were presented in-depth in the previous sections. To showcase the wide range of applications this simulator can be used for, here we present a few examples by conducting illustrative experiments. In Section 2.1, the general settings of the experiments are described and the results are discussed in Section 2.2. Some sanity checks of the simulator with a rudimentary population structure are also provided in the S3 File for further reference.

### 2.1 General experimental setup

In order to assess the simulation using a real-world scenario, the simulator requires a sample population structure, defining the primary properties of the population and the attributes of the disease that is planned to spread through the population. This data is provided to the software by two configuration files, one containing the population settings and the other containing the disease attributes. Prior to running the simulation engine, the configuration files need to be prepared.

For our experiments, a configuration file for the population generator is designed based on the structure of a small town's population. Likewise, a configuration file is prepared based on the known attributes of COVID-19 as an exemplary infectious disease. Before the simulation starts, the software parses these configuration files and constructs the population based on the retrieved information.

**Table 5. Set of family patterns and the probability of their occurrence (M: male, F: female, {}: A family gender pattern).** Any other arbitrary family pattern can be easily defined in Pyfectious. For brevity, the age and the health condition of the family members are excluded from this table. They are sampled from a truncated normal distribution for every family member.

| Family Size | Genders | Probability |
|---|---|---|
| 2 | {M, F} | 0.21 |
| 3 | {M, F, M}, {M, F, F} | 0.3 |
| 4 | {M, F, M, F}, {M, M, M, F} | 0.19 |
| 1 | {F}, {M} | 0.126 |
| 5 | {M, F, M, F, F} | 0.124 |
| 6 | {M, M, M, F, F, F} | 0.05 |

This section is dedicated to explaining the properties involved in the configuration files, along with a concise justification of the design procedure.

**2.1.1 Population generator.** Designing a representative population structure is critical for the simulation to generate reasonable results that bear a resemblance to the available real-world statistics of the pandemic. Thus, the configuration file for the population generator must be designed carefully and based on realistic assumptions. We use the information provided by reputable population census and statistics centers. In particular, we use two primary sources of information to adjust the structure of the generated population. First, the United States Census Bureau's tables and data of the US population [28]. Second, the Eurostat [29] data collection that provides information on the demographics of a number of European countries.

The population generator configuration file comprises the following items.

1. `Population Size`: This attribute determines the total size of the population and is set to 20,000 in our experiments, which is roughly the average population of a small town.

2. `Family Patterns`: A list of family patterns and the probability of their existence in society are required to disperse the population among the families. Furthermore, as discussed in Section 1.1, a family pattern enables the simulation to assign individual attributes, for instance, gender and health condition, to the people. Six family patterns are designed and reported in Table 5 for our intended experiments.

3. `Community Types`: As illustrated in Section 1.1, after distributing the individuals of the population to families based on the provided family patterns, the individuals are also assigned to communities that are the closest match to their personal attributes and the attributes of the family they belong to. The conducted experiments in this section are based on the communities briefly described in Table 6.

4. `Distance Function`: A function that determines if two individuals are in close contact. Here, the Euclidean distance is used as a measure of proximity between two individuals. Other measures can be employed depending on real-world circumstances.

**2.1.2 Disease properties.** The disease configuration file contains the fundamental attributes of an infectious disease. Here, we explain these attributes and the reasoning behind the selected value for each of them.

1. `Infection Rate`: Sampled from a uniform distribution whose support is determined based on real-world reports. We adjusted the support of the distribution, i.e., Uniform[0.1,

**Table 6. The design of the communities and subcommunities for an exemplary city (society).**

| Community Type Name | Number of Communities | Sub-Community Types | Transmission Potential | Connectivity Matrix |
|---|---|---|---|---|
| School | 40 | Teacher Student | Very High | High Density |
| Workspace (Medium) | 800 | Worker Potential Client | Very High | Medium Density |
| Workspace (Large) | 50 | Worker Potential Client | Moderate to High | Medium Sparsity |
| Gym | 50 | Trainer Client | Moderate to High | Medium Density |
| Public Transportation | 10 | Commuter Staff | Low to Moderate | High Sparsity |
| Restaurant | 80 | Staff Costumer | Moderate to High | Medium Density |
| Cinema | 15 | Staff Costumer | Low to Moderate | Medium Sparsity |
| Mall | 5 | Staff Costumer | Very Low | High Sparsity |
| Hospital | 2 | Healthcare staff Patient | Very Low | Medium Sparsity |

0.6], in the experiments based on the data available from Wuhan [30], where there were less restrictive measures at the beginning.

2. `Immunity`: Based on the previous studies on COVID-19, e.g., [31], no prior immunity has been detected in most of the studied cases. Therefore, the immunity against infection for the first time must be sampled from a distribution with support relatively close to zero. However, once a person recovers from the disease, the possibility of reinfection in the short term is infinitesimal. The immunity parameter is sampled from a distribution that is designed based on this assumption: A person has a negligible immunity against the virus in the first infection, and the immunity is exponentially raised upon each infection.

3. `Incubation Period`: We consider a truncated normal distribution (mean = 5.44, 95%-CI (Confidence Interval) = (4.98, 5.99), STD = 2.3) for the duration of the incubation period based on the reported values in the data collected by [32] and more notably the study conducted by [33].

4. `Disease Period`: This parameter is sampled from a truncated normal distribution (mean = 6.25, 95%-CI = (5.09, 7.51), STD = 3.72) based on the values reported in [33, 34].

5. `Death Probability`: The value of death probability, also known as mortality rate, is available through the hospital's statistics for patients who are diagnosed with COVID-19. Here, the death probability is sampled from a truncated normal distribution (mean = 1.4%, 95%-CI = (1.1%, 1.6%), STD = 2.4) as reported in [35, 36].

## 2.2 Evaluation and assessment of the results

To illustrate the simulator's potential and applications, we have conducted several experiments, ranging from exploring the outcome of changing the disease parameters to imposing restrictions to control the disease's spread. The conducted experiments are based on the configurations described in Sections 2.1.1 and 2.1.2.

We deployed our experiments on a cluster distributed on a multitude of computational nodes to accelerate using parallel execution. Every particular curve that we obtained through our experiments is produced by 32 simultaneous executions combined in a single graph. We use the moving average, with a window size of 2, to reduce the effect of high-frequency oscillations and improve the visibility of the trend. The variations are typically caused by the chosen sampling rate, which is 7 hours in our experiments. Comparing our observations in Fig 8b with the statistics published by the authorities, for instance, the US COVID-19 statistics [37],

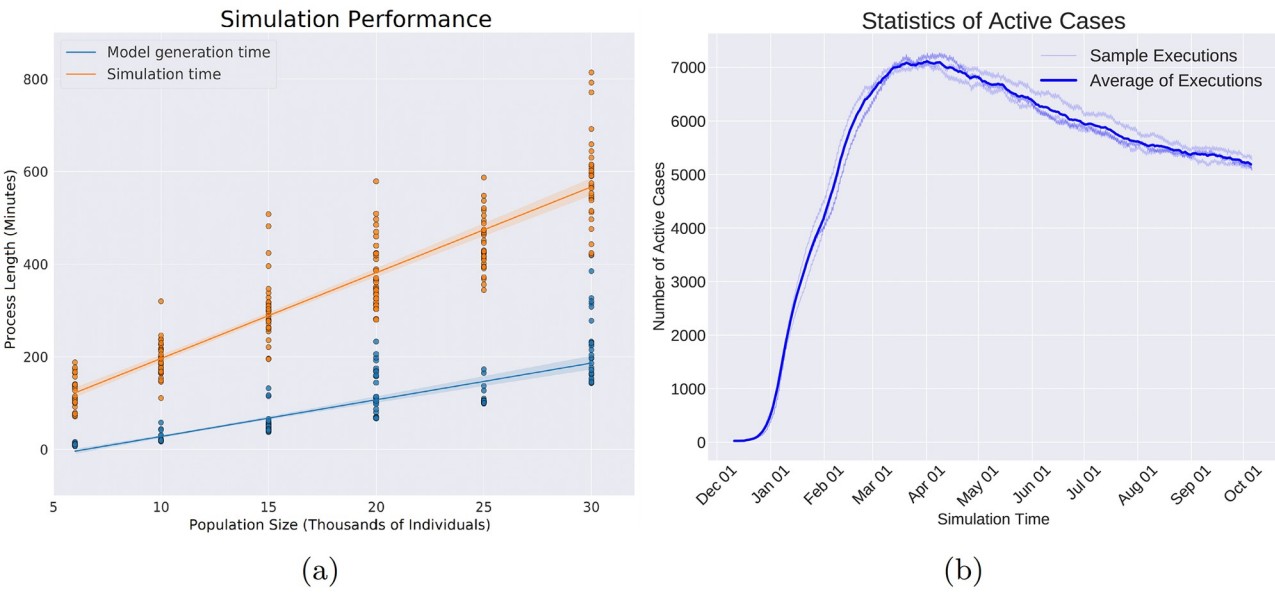

**Fig 8.** a) The time it takes to generate the population and to simulate the propagation of the disease for 48 hours is plotted for six cities with population sizes from 6k to 30k. The clock (virus spread period) is 60 minutes in this experiment. The horizontal axis represents the population size, and the vertical axis represents the total process time. The experiment for every size of the population is repeated multiple times (each vertically aligned dot corresponds to an experiment) to achieve confidence, and the straight line indicates the trend. b) The number of active cases versus time is shown for a sample city with a population size of 20k. To emphasize the probabilistic nature of Pyfectious, the same experiment is repeated multiple times, and the effect of this randomness is seen by observing slightly different trajectories. The blue curve is the moving average (window size of 2) of all executions to show the trend.

we conclude that the observed oscillations are not unexpected and do not affect the trend of the simulation graphs.

To deliberately evaluate the experiments, they are divided into five categories, each explained in the subsequent sections.

**2.2.1 Performance.**   In this section, we discuss practical and theoretical issues related to the performance of the simulation software Pyfectious.

**Theoretical performance measures**. The dominant computational demanding component of most simulators is the virus spread parts, since they grow as $\Omega(M^2 S)$, $M$ being the population size and $S$ being the number of virus spread events. This is clear as, within a complete graph with $M$ nodes, $M^2$ edges exist and $S$ iterations totally occur over these $M^2$ edges. Pyfectious is no exception to this rule. Indeed, employing an event-based method, Pyfectious has the computational complexity of $\Theta(M^2 CS)$ for virus spread, where $C$ is the average connectivity level of the underlying graphs. Moreover, the computational complexity of the other dynamics (planning the daily schedule and other transition events) of the Pyfectious is $\Theta(MS)$. Hence, the computational complexity of Pyfectious is theoretically optimal being $\Theta(M^2 CS)$ overall.

**Empirical performance measures**. The first experiment in this section, shown in Fig 8a, aims at measuring the required time for the population generation and simulation phases. Since the generated models may be saved and reused when running the simulation more than once, its required time can be amortized when simulating the same model multiple times or with different disease attributes. Evidently, the experiment proves an empirical linear relationship between the population size and both the simulation and population generation times. This result is promising and shows the scalability of Pyfectious for simulating larger cities with

more complex population structures. This empirical evaluation is consistent with the theoretical analysis as in real-world settings $\Theta(M'C) \approx \mathcal{O}(1)$ is usually constant for sufficiently large ($M'$) communities.

**The case of multi-resolution simulation**. As stated in the previous paragraphs, the computational cost of simulation is $\Theta(M^2CS)$ which is equal to $\Theta(M^2Ct/T)$, where $t$ and $T$ are simulation time and clock period respectively. Thus, the computational cost is linearly proportional to the system's resolution $1/T$ (inverse of the clock period). Standardizing the disease transmission probability discussed in Section 1.2.4 and uniformly selecting virus spreads between each interval addressed in Section 1.2.4 guarantee almost invariant simulation results with respect to the system's resolution in an irreducible (a stochastic process in which every state is reachable by other states) environment.

The reason for that is the unvarying nature of disease transition probability with respect to the system's clock period. Unfortunately, this irreducibility is not satisfied in disease propagation simulations because of a few reasons; for example, after getting infected, the patient either dies or becomes immune. Despite this, almost invariant behaviors are observed in the empirical results in Fig 9a. Results in relatively low-resolution regimes (clock period $T = 320$ mins) are approximately consistent with a higher-resolution regime (clock period $T = 80$ mins). Hence, we can gain $320/80 = 4\times$ speedup with almost the same conclusion in this experiment. However, as depicted in Fig 9b, in a very-low-resolution regime (clock period $T = 1080$ mins), the reducibility of the simulation process impedes this unchanging behavior and may cause radical changes in the qualitative characteristics of the simulation curves. A similar plot but with a smaller population size is depicted in S4 File. All in all, the system's resolution is adjustable according to the user's computational resources, while preserving some aspects of the results.

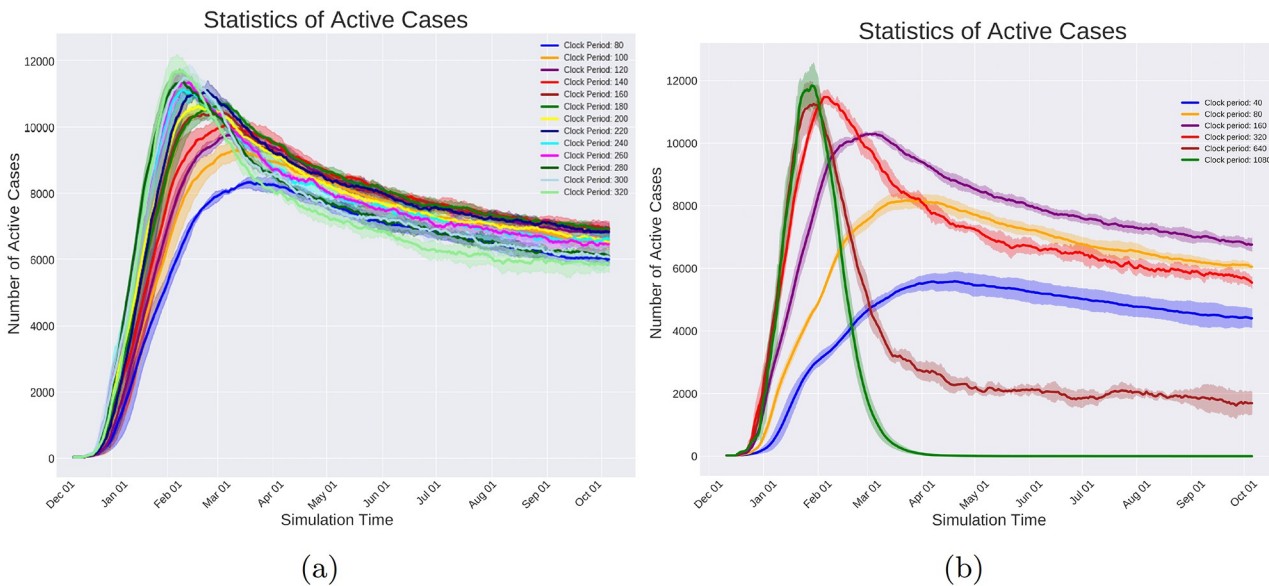

(a) (b)

**Fig 9.** Experiments demonstrated here are focused on the almost invariant behavior of the simulator within a reasonable chance of the temporal resolution either when applying various control policies or by changing the spread period. The population structure and other details related to these experiments are the same as Section 2.1, with a population size of 20k people. (a) Shows this consistency for clock periods in the interval [80, 320], whereas (b) suggests that the reducibility of the system is a barrier that breaks this invariance feature in case of a radical reduction of temporal resolution.

**Table 7. Performance measure of commands.** Runtime for commands is averaged over 100 command executions, showing the insignificance of commands runtime compared to other components of the simulator.

| Command | Time (ms) | Integrated Condition |
|---|---|---|
| Quarantine_Single_Community | 236.7(±37.2) | Time_Point_Condition |
| Quarantine_Single_Family | 64.2(±11.3) | Time_Period_Condition |
| Quarantine_Multiple_Communities | 562.5(±43.6) | Time Period Condition |
| Quarantine_All_People | 622.8(±56.1) | Time Point Condition |
| Quarantine_Diseased_People | 641.4(±67.8) | Statistical Family Condition |

**Computational overhead of control measures**. The induced overhead of control measures varies depending on the corresponding conditions and commands. For example, `Time Point Condition` applies only once but `Statistical Ratio Condition` can be applied at most $t/T$ times. We ensure that all of the previously provided conditions and commands are implemented optimally by using dynamic programming and proper indexing techniques. Each condition is satisfied at most $t/T$ times and each command execution takes at most $\mathcal{O}(M^2C)$ which means the overall overhead of a control measure is at most $\mathcal{O}(M^2Ct/T)$. However, the complexity of most of the control measures provided in this paper such as the experiments in Section 2.2.4, where $m$ is the size of the target community. A practical demonstration of the computational overhead of control measures is given in Table 7.

**2.2.2 Sanity checks.** In this section, we present primary experiments related to the soundness of the simulation software Pyfectious.

**Sanity checks**. In the experiment whose result is shown in Figs 8b and 10a, the graphs of the number of active cases versus time are reported for six executions. The oscillations caused by sudden changes in the active cases' statistics are observable, especially near the curve's

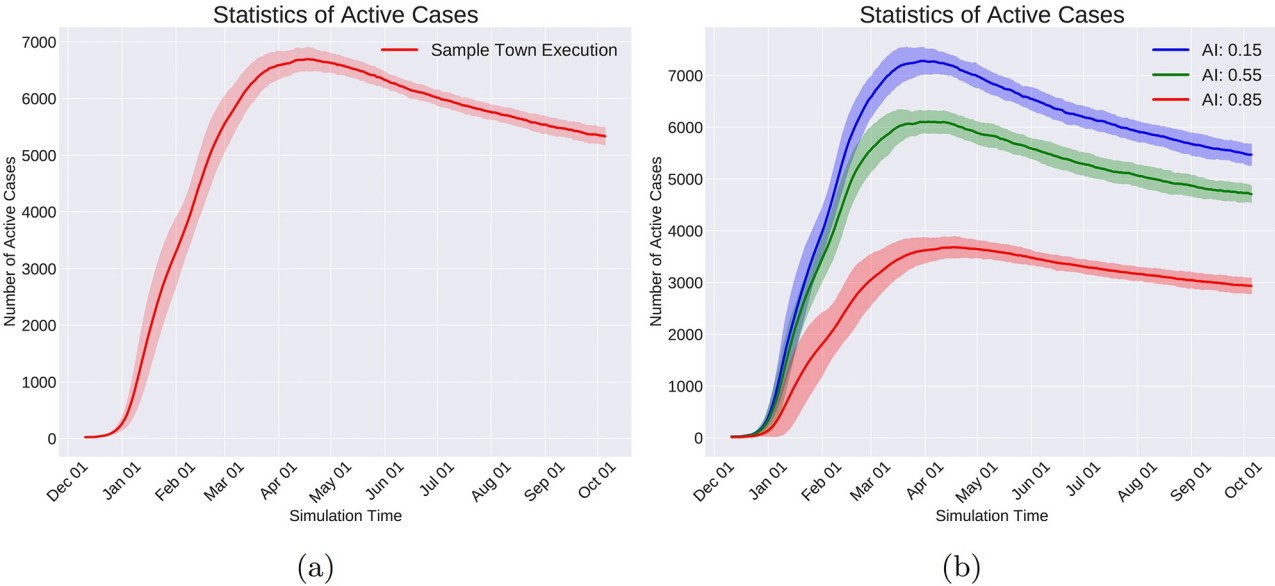

(a)                                                          (b)

**Fig 10.** a) The simulation is executed without any control measure, and the number of infected individuals is plotted versus time for the period of 10 months. The halo around the solid curve is the confidence interval obtained by multiple runs. At each round, the parameters of the population and the disease are re-sampled from the specified distributions. b) The curves of the number of active cases versus time are plotted for different immunity rates. The immunity rate is sampled from three uniform distributions with different mean values. As can be seen, more significant immunity rates give rise to flatter curves. Note that AI initials in the legend stand for Average Immunity, which is the mean of the uniform distribution from which each curve's immunity rate is sampled.

global maximum, where there is the largest number of active cases. As mentioned before, the moving average of these curves is shown to illustrate the trend better. Notice that a slight difference between the produced trajectories in Fig 8b is expected due to the probabilistic nature of Pyfectious at multiple levels. Numerous sampled trajectories similar to those shown in Fig 8b form a halo around the average trajectory as in Fig 10a to obtain confidence in the results. Every experiment in this paper is executed multiple times to obtain such confidence intervals and be robust against randomness artifacts.

**Discussion on herd immunity**. Although no control measure is applied during the simulation in Fig 10a, the number of active cases declines after reaching a certain level. This incident, called herd immunity [38], is aligned with the course of infectious diseases in the real world. It shows a reduction in the number of active cases after a certain fraction of the population has been infected by the virus and developed a level of immunity. We implemented an immunity model based on the available real-world information. Inspired by [39], our model incorporates a minor chance of reinfection (about 1%), which significantly drops after the second infection. Aligned with real-world experiences, a decline occurs in the number of cases after a sufficient portion of the population (about 70–80% in our simulation) recovers from the disease.

**2.2.3 Changing the attributes of the disease.** To investigate the effect of changing the disease properties, this section covers a comprehensive study on how changing disease attributes affect its spread through a structured population.

**Immunity variations**. To assess the influence of immunity distribution on the results, we organize three sets of experiments, depicted in Fig 10b. Each curve shows the evolution of the number of active cases where the immunity rate is sampled from a uniform distribution with a specified range. In the lowest immunity level, a larger population contracts the disease as the population has the least resistance against the infection. Notably, increasing the immunity level to intermediate and high results in lowering the peak of the active cases' curve. For instance, if a fraction of the population is vaccinated, the immunity rises within that group, and the trajectory of active cases reorients from the blue curve to the red or the green one. This incident is often referred to as "flattening the curve."

**Modifying the infection rate**. In this experiment, we investigate the effect of infection rate on the disease's spread in the population. To obtain confidence in the results, each experiment is executed 32 times with the infection rate that is sampled from a uniform distribution with a specified range. The curves in Fig 11a indicate the results for five non-overlapping uniform distributions from which the infection rate is sampled. It can be seen that for the exceptionally high rate of infection (i.e., the infection rate is sampled from a uniform distribution whose support occupies larger values), the number of active cases increases with a significant slope before reaching the peak of the curve. As the area under the curve of active cases shows in Fig 11a, the total number of infected people during the epidemic increases with an increase in the infection rate.

**Initially infected cases**. The first set of infected individuals in a population plays a crucial role in spreading the disease. Here, we check this effect by studying the course of the epidemic when the number of initially infected people is set to either 6 or 25 with the added information that the smaller set is chosen from the large workspaces and schools. Again, we run every experiment multiple times to ensure the results are not by accident. As Fig 11b shows, when only six infected people exist at the beginning of the simulation, the error bounds are wider compared to when there are 25 initially infected people. This effect is expected because, for a more significant number of initially infected people that are randomly assigned to communities, most of the communities will have at least one infectious member. Hence, there will be minor variations across the executions of the simulation compared to the situation when a few individuals are chosen from a different set of communities at each execution. The other

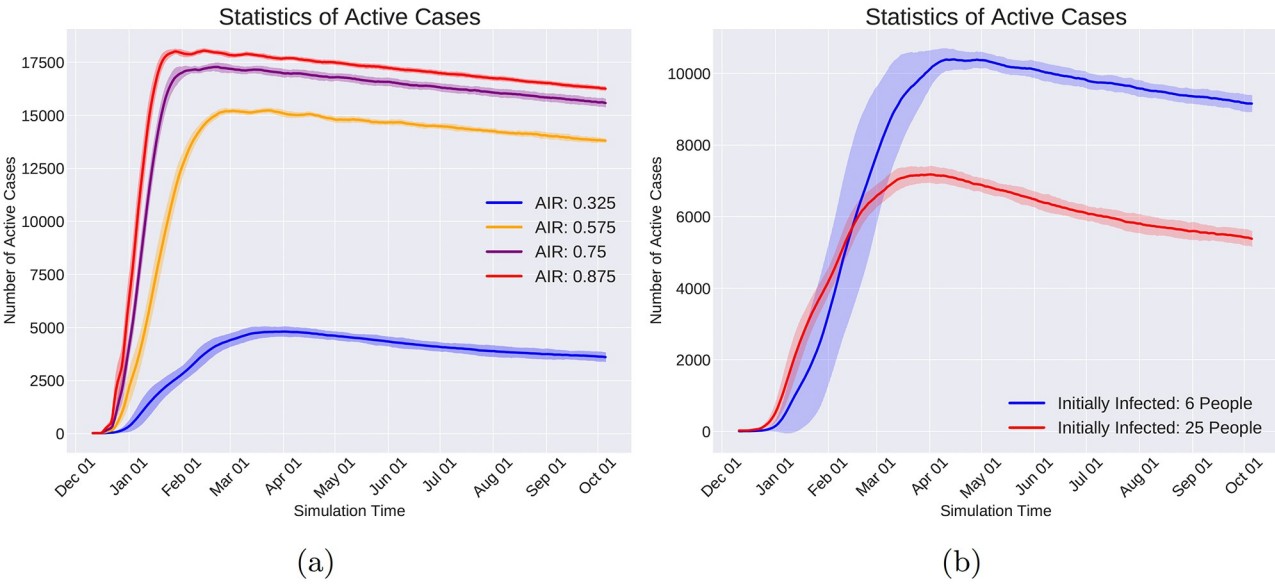

**Fig 11.** a) The number of currently infected individuals is plotted versus time for different values of the infection rates. The infection rates are sampled from uniform distributions with different mean values. It is observed that a larger infection rate increases the slope of the curve which means a faster spread of the disease early after the advent of the outbreak. As a result, it takes less time for the number of active cases to reach its peak. Notice that AIR stands for the Average Infection Rate, which is the mean of the distribution from which the infection rate is sampled. b) The spread of the infection is shown versus time for different numbers of initial spreaders where the smaller set is chosen from communities such as large workspaces and schools that are suitable places for infecting many people. The confidence intervals are expectedly wider for a smaller initially infected set because it results in some communities without an initial spreader and consequently a less homogeneous spread of the disease. The observation that the peak of the graph with a smaller initial set is higher than the one with a larger initial set emphasizes the hypothesis that some roles and places need special treatment early in an epidemic even though only a few of their individuals can be initially infected.

notable observation is that even though the blue curve initiates from fewer infected people, it shows a larger set of infected people eventually. The reason is that its initial set is intentionally chosen from communities, such as schools or large workspaces, making it easier to spread the disease throughout a large population in a short time.

**Incubation and disease period**. As already discussed in Section 2.1.2, incubation and disease period parameters define the temporal behavior of the infectious disease. As Fig 12a shows, changing these parameters has a significant effect on the curve of active cases. For instance, increasing the incubation period creates a longer flat curve at the beginning of the outbreak. It can also be seen that an elongated incubation period shifts the curve forward in time, with an almost equal peak of the number of active cases. Simultaneously, a shortened length of the disease period decreases the peak and the total cumulative number of active cases. This result is expected because a longer period of disease without a strict control measure to keep the infected individuals away from others increases the chance of spreading the infection.

**2.2.4 Applying control measures.** As introduced in Section 1.2.5, Pyfectious offers a straightforward and flexible way to impose control measures during the course of the simulation to emulate real-world epidemic containment policies. The control policies can appear in numerous forms, including quarantining people or communities and reducing the number of people in some sectors of society. Here, we present a couple of experiments to illustrate the effect of control measures that are similar to those applied in the real world. Typically, there is a trade-off between the strength of the control measure and its outcome. The most strict measures, such as forcing everyone to stay at home and in isolation from other family members, stop the spread of the disease but entails enormous economic and societal costs. Pyfectious

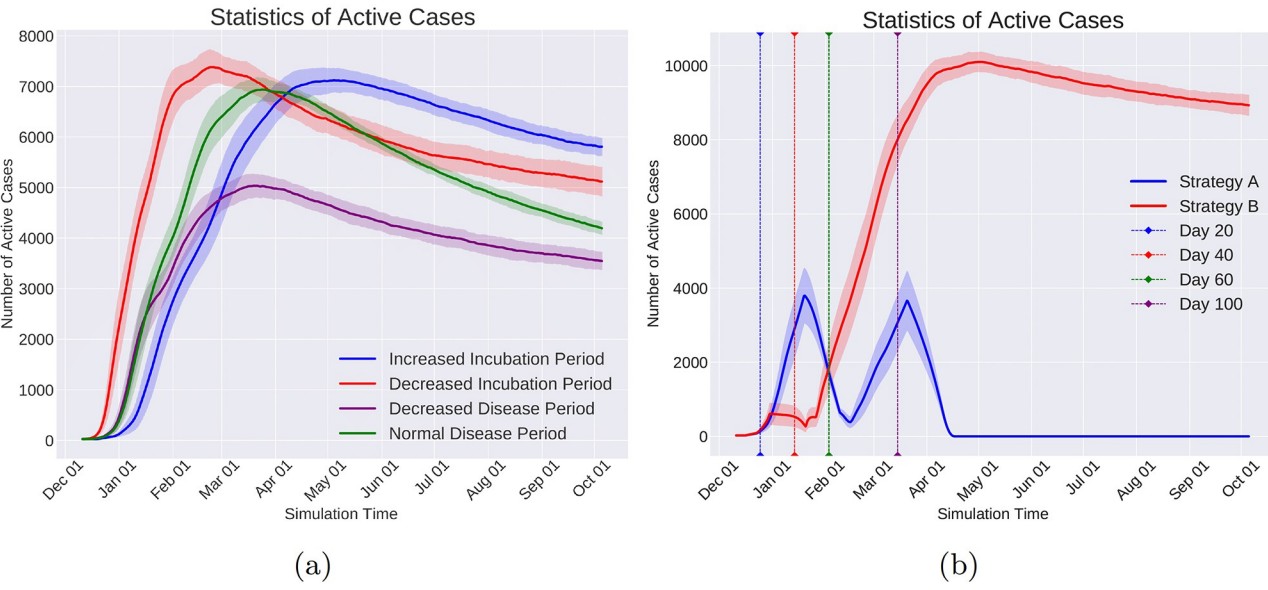

**Fig 12.** a) The effect of the length of the incubation period (the period in which the infection is not detectable) and the disease period (the period in which the individual is infectious) is shown by changing these parameters of the disease. The curves correspond to a normal incubation and disease period, an increased incubation period by 3.5 days, a decreased incubation period by 3.5 days, and a decreased disease period by seven days. b) The outcome of two quarantine strategies. Strategy A: Enforce a quarantine 40 days after the outbreak, lift it after 20 days and enforce it again after 40 days. Strategy B: Enforce a quarantine 20 days after the outbreak and lift it 20 days later. The oscillatory curve is expected as the remaining active cases after the initial quarantine will be the initial spreaders for the next wave of the epidemic.

allows us to investigate this trade-off by changing the strength of the control measures in an almost continuous way to find the optimal restrictive rules with a reasonable cost. In the following, some of the control measures inspired by real-world policies are tested.

**Full quarantine**. This policy, whose result is shown in Fig 13a refers to the most strict quarantine method isolating every discovered infected individual. Each curve of Fig 13a represents the effect of applying the strict full quarantine with different starting dates. As the quarantined infected people are no longer able to infect others, the complete isolation causes a sharp drop in the number of active cases until the epidemic eventually vanishes. As expected, it is clear that the full quarantine is most effective if it is applied as early as possible after the outbreak of the infection.

**Enforce and remove a quarantine**. In Fig 12b, the effect of enforcing a quarantine early and removing it after some time is presented. As appears in the results, an early quarantine could be effective if placed and removed at particular time points (as in Strategy A). However, it could also fail to contract the virus if not appropriately planned (as in Strategy B), i.e., the start and termination time are not sufficient to control the spread, and the pandemic strikes back after removing the quarantine.

**Partial quarantine**. As opposed to the full quarantine measures, in Fig 13b, we study the effect of quarantine under a more realistic assumption, that is, the detection of the infected people is not absolutely accurate. For instance, a 50% error indicates that the detection and testing mechanisms are cables of detecting only half of the infected population. Once infected people are detected, they are treated with total isolation as in the full quarantine policy. Nevertheless, complete isolation is still effective in controlling the epidemic even when the detection rate is significantly lower, e.g., only 20% of infections are detected.

**Quarantining specific sectors**. This control measure restricts specific communities or subcommunities, such as gyms, public transportation, and schools. The result of shutting down

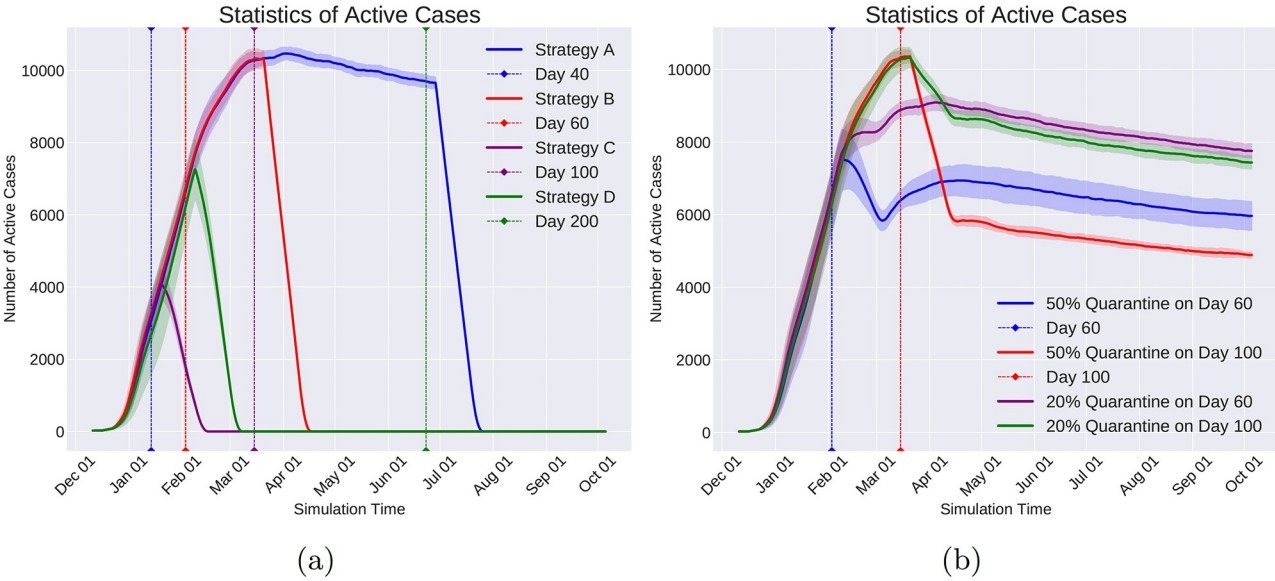

**Fig 13.** a) This experiment focuses on enforcing universal quarantines (isolating every infected individual after detection) on a specific day after the outbreak. Here, the quarantines are applied both before and after the day when the curve of the active cases reaches its peak. Strategies A, B, C, and D enforce a quarantine at 200, 100, 40, and 60 days after the outbreak, respectively. b) This experiment studies the effect of partial quarantine where a specified ratio of currently infected individuals are isolated at a specified date (i.e., the control measure is triggered by a time point condition). The partial quarantine represents a real-world scenario where there is uncertainty in detecting the infected individuals, which can be caused by numerous reasons such as inaccurate test kits or individuals with mild symptoms that do not visit hospitals or test facilities.

some communities is shown in Fig 14a. We study the effect of closing gyms, restaurants, cinemas, malls, and public transportation. As a result, we observe that the most effective decision is to impose a general lockdown on workspaces. It can be seen that closing public transportation and shopping malls have a negligible impact on flattening the epidemic curve. This seemingly counter-intuitive observation is justified, as people will end up in close contact with the infected individuals at their destinations, regardless of how they get there. Moreover, due to the size of the considered city, the passengers spend a short period in public transport facilities, which decreases the chance of getting infected.

**Restricting specific roles**. This class of control measures, also known as working from home (WFH) policies, concerns restricting the physical presence of the employees of specified jobs whose physical presence is not absolutely necessary. It was described in Section 1.2.1 that, in Pyfectious and inspired by the real-world population structure, each community (e.g., schools) can have multiple roles (e.g., teachers, students, staff) with their special daily schedule. This control measure is especially effective and less costly because workspaces are critical hubs in the spread of the disease, and many jobs can be performed remotely thanks to the developed online communications in many areas. The results in Fig 14b show a decline in the active cases a couple of days after the restrictions are enforced. It is observed that the ratio of the isolated employee plays an essential role in the outcome. In our experimented setting, a restriction that involves only 30% of the working force seems ineffective compared to a situation where 60% of the employees are working from home.

Even though a diverse set of control measures are introduced above, any arbitrary customized control measure can be defined thanks to the Condition/Command framework we introduced in Sections 1.2.5 and 1.2.5. Here is an imaginary, complicated example to elucidate the level of flexibility: A user may create a specific condition that is triggered whenever one-third of the friends of a person who shares at least half of their communities are tested positive

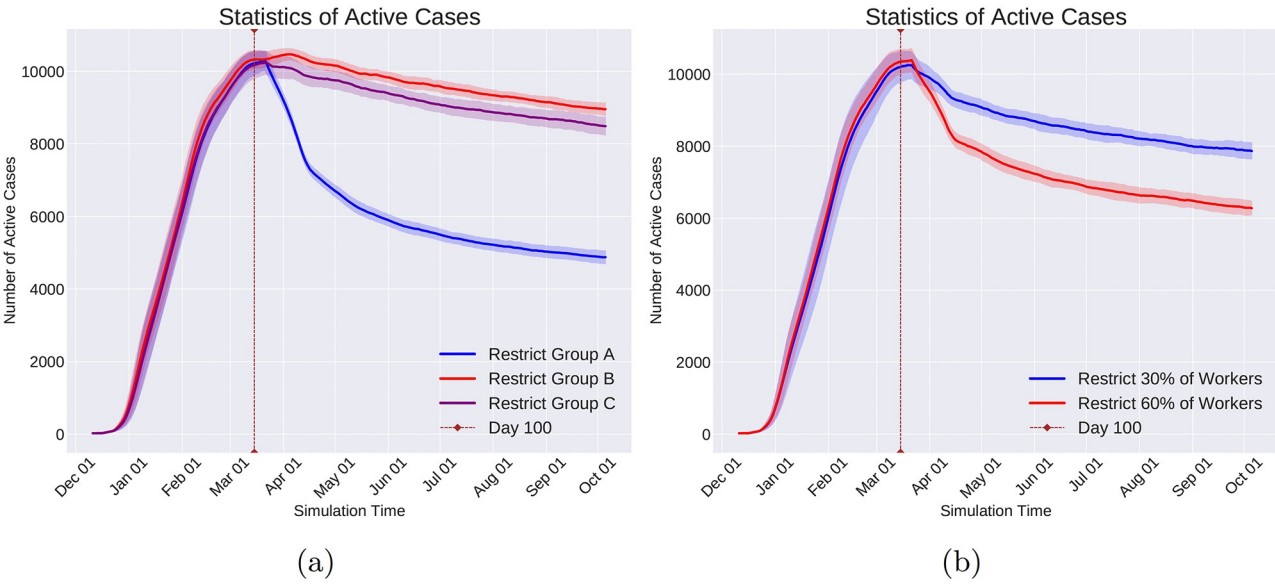

**Fig 14.** a) This graph shows the effect of control policies that target specific sectors of society. Each curve corresponds to shutting down a different place. Group A includes all workplaces of any size. Group B consists of gyms, restaurants, and cinemas. Group C includes more public places such as malls and public transportation. b) This graph shows the effectiveness of quarantining specific roles (subcommunities) in society. The curves show the spread of the infection when different ratios of workers of any kind are quarantined. The effect is expectedly significant because the workers spend so much time in their workplaces every day, and many individuals often visit a workplace during working hours, which makes it a suitable place for spreading the infection.

for the disease. In addition, a specific command can also be defined that causes the members of the aforementioned friends' households to be banned from using public transportation. By integrating these arbitrary conditions and commands, the user has defined a customized intervention. In other words, defining new and augmenting the existing control measures are part of the design principles in Pyfectious, a property that is unparalleled in most of the existing simulators with this level of detail.

**2.2.5 Closed-loop policies.** The experimented policies in the previous sections did not automatically react to the changes in the epidemic condition in the population. A more innovative policy should be able to adjust its commands when the conditions change. Pyfectious is equipped with special objects that constantly monitor the population and fire a trigger signal when a pre-specified condition is met. As discussed in Section 1.2.5, the trigger signal of condition objects can be fed to either an observer object to record the statistics or a command object to issue a new restrictive rule or relieve the existing ones based on the current state of the epidemic. This closes the loop between command and observation and renders a closed-loop policy. To illustrate closed-loop policies, a simple controller is designed and is triggered when a condition is satisfied. Here, we define the condition as the moment when a ratio of two statistics from the population surpasses a specified threshold. These experiments also illustrate the substantial flexibility of the simulator to assess unlimited scenarios for epidemic control.

**Simple cut-off mechanism**. Viewing the entire population as a dynamic system, this policy acts as a step controller [40] that gets activated when more than 10% of the population are infected (see Fig 15a). The ratio threshold implies the tolerance of the policymakers and may be imposed by economic and societal factors.

**Bang-bang controller**. This policy, called bang-bang controller, switches between two defined states: enforce and release the quarantine. As depicted in Fig 15b, this strategy keeps the ratio of active cases to the population size between 0.1 and 0.15 until the disease is no

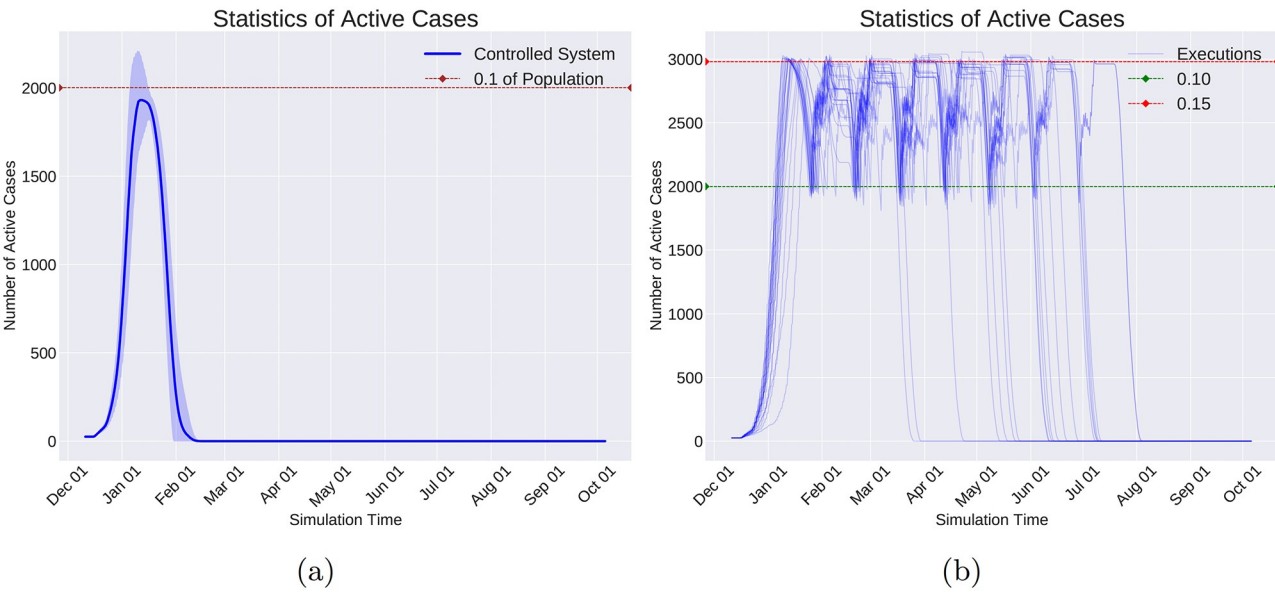

(a)　　　　　　　　　　　　　　　(b)

**Fig 15.** a) In this experiment, a command is set to quarantine the infected individuals in the population when 10% of the whole population is infected (a ratio condition triggers the control command). b) In this experiment, infected individuals are quarantined when more than 15% of the population are infected, and the quarantine is lifted when the ratio of infected individuals drops below 10%. In the terminology of control theory, this strategy is known as a bang-bang controller.

longer capable of spreading, i.e., herd immunity is reached. This scenario is specifically important since it reduces the burden on the healthcare systems by keeping the active cases below the capacity of the hospitals and, at the same time, controls the financial burden of a long-term comprehensive lockdown. Multiple executions are plotted in Fig 15b whose oscillations between two thresholds depict the actions of the bang-bang controller. It can be seen that based on the initial condition of the simulation and the initially infected individuals, some curves start dropping earlier than others. However, all dropping curves share the same slope from the point when herd immunity occurs and no new individual gets infected.

**2.2.6 Finding an optimal policy.** As mentioned earlier in the Introduction, in addition to predicting the course of an epidemic under different individual-level control policies, the ultimate goal of Pyfectious is to discover smart and detailed policies that might be hard for human experts to find. The purpose of this experiment is to showcase this feature and find the most effective policy that minimizes the negative impacts of an epidemic of a specified disease on a specified population. In the real world, the outcome of the employed control measures in previous epidemics combined with the knowledge of the epidemiologists is used to devise control strategies to contain the spread of a novel infection. However, the complexity of the problem grows so quickly that the predicted outcome becomes unreliable even if there is a slight difference between the current and the previously experienced conditions. Pyfectious as a detailed and high-performance simulator gives the possibility to test many proposed control strategies quickly and accurately. To find the best control measure, we offer an automatic method that cleverly searches in the space of fine-grained policies to approach the one that performs best in the specified population.

The experiment follows the steps below.

1. Constructing the structure of the population of interest: We employ the structure the same way as the previous experiments, with the exception that the population size is one-tenth.

Since the population structure, i.e., communities and family patterns, is unchanged, the results are still practical.

2. Defining a cost function: We pick the maximum height of the curve of the active case as the cost function. Then, the goal will be finding the control measure during the entire course of the epidemic such that the number of active cases never gets so large at any time. This objective function assures that the health system will not saturate.

3. Determining the optimization parameters: Every optimization is done concerning a set of parameters. As we search in the space of control measures, the control policy needs to be encoded in a few numerical parameters. In our experiment, the control measure consists of restricting students, workers, and customers. The ratio of the restricted fraction of each community is a controllable variable denoted by $\alpha$, $\beta$, and $\gamma$, respectively, for instance, in the case where $\alpha = 0.8$, only 20% of students are permitted to attend the schools physically. Hence, searching for an optimal policy amounts to finding the best values for these parameters.

4. Defining the economic constraints: Searching for the optimal control measure is essentially an optimal control problem, where the control action always comes with some cost. Moreover, in order to avoid picking a trivial solution, we have to introduce a set of constraints on the restriction ratios. For instance, setting all restriction ratios to 1 is clearly the most effective and yet, the most expensive solution. We introduce some constraints to eliminate the trivial solutions. To do so, the sum of the restricted proportion of the roles is constrained as $\alpha + \beta + \gamma = 1.4$. Moreover, the ratio of each role that can be isolated is also upper bounded as $0 < \alpha < 0.7$, $0 < \beta < 0.7$, and $0 < \gamma < 1$, implying that some roles cannot be completely remote. In the absence of these constraints, the trivial solution would be isolating the 100% ratio of all three considered roles.

5. Selecting an optimization algorithm: After defining the above-mentioned components of the problem, various non-gradient-based methods can be employed to carry out the optimization. It is clear that gradient-based methods cannot be used here because the gradients need to be back-propagated through the entire simulator which is not differentiable. Viewing the population and disease as the environment and control commands as the policies, the discovery of the optimal epidemic control policy can be seen as a reinforcement learning problem. We postpone further elaboration on this application of Pyfectious to future work. Here, a simple probabilistic search method (Bayesian optimization) is employed that can be regarded as a special subset of reinforcement learning algorithms. We use Hyperopt [41] package to find the optimal set of restriction ratios based on the predetermined criteria. Hyperopt exploits the Tree of Parzen Estimator (TPE) algorithm, explained in [41], to find the set of parameters corresponding to the most significant expected improvement at each iteration.

**Discussion on optimization results**. The course of the optimization process is shown in Fig 16. The found optimal restriction rule is to enforce 60% of students and workers to stay home. The restriction is less severe for the customers of unnecessary activities where only 20% of their population are to be restricted. The discovered policy is aligned with our experience of real-world scenarios, where schools and workspaces have the highest risk for spreading the infection since people are in close contact for a relatively long period per day. More importantly, based on the constraints mentioned in item 4 of Section 2.2.6, maximally 70% of students and workers can be restricted while this ratio is unbounded for customers. As seen in Table 8, it turns out that, although workers and students are considered the groups with the

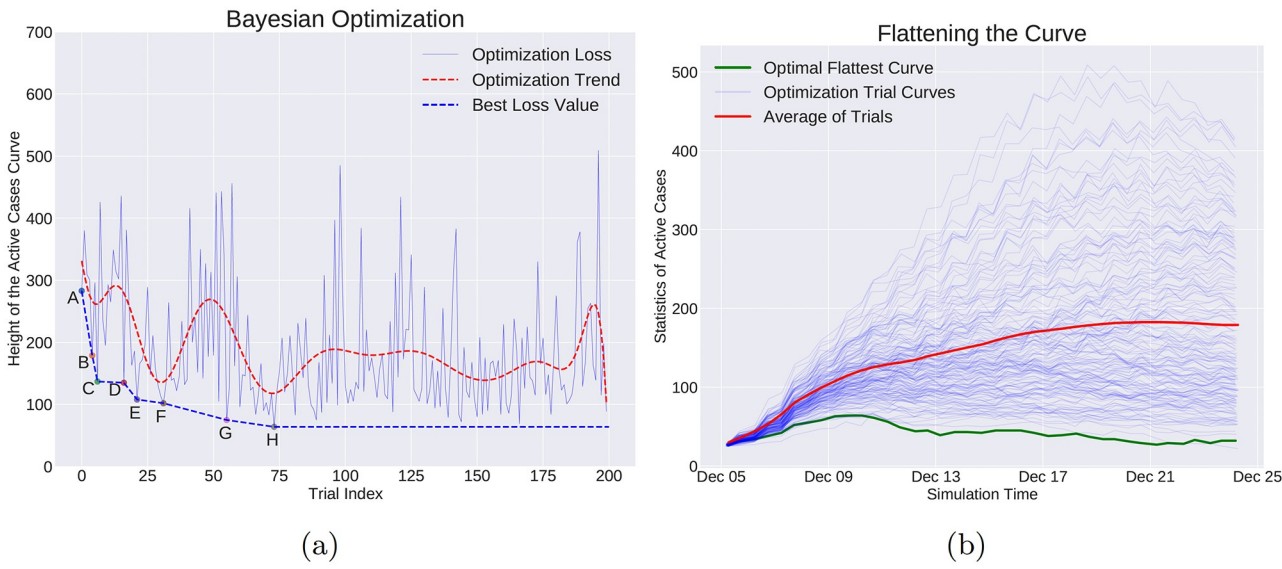

**Fig 16.** a) The agent aims to minimize the loss function defined as the peak of the active cases. The optimization variables are the ratio of three roles that must be quarantined, and the ratios are constrained to be bounded from above and sum up to a constant value. The upper bound constraints are placed to take into account the cost of shutting down the economy and the trivial solution that is quarantining all individuals. The graph shows the result for 200 trials. The blue dashed line is the lower envelope of the cost produced by the discovered solution at every trial. Each point from A to H corresponds to the minimum cost up to that trial. The discovered policy associated with each of these points can be seen in Table 8. b) The curves that show the number of active cases versus time for each round of the optimization is plotted in this figure. These are actually the curves we need to flatten to protect the healthcare system against overloading. It can be seen that the discovered strategy with the least cost corresponds to the flattest curve. (The population size is reduced by a scale of 10 to boost the computation time.)

highest spreading potential, imposing the maximum possible restriction on these groups is not the most effective policy. Instead, and in light of the economic constraint $\alpha + \beta + \gamma = 1.4$, the algorithm learns to restrict 60% of the population of students and workers and leave the other 20% of the restriction budget for customers. This solution is relatively counter-intuitive, which emphasizes the benefit of Pyfectious to find control strategies that might be difficult to find by human experts.

**Table 8. A summary of the discovered control strategies during the rounds of the optimization process.** The first column is the index of the discovered quarantine strategy. Each strategy consists of three ratios that show the portion of each group of {students, workers, customers} to put in quarantine. The second column indicates the iteration at which the associated strategy is found, and the rightmost column shows the value of the cost function for that strategy. This table only incorporates the iterations at which the agent improves the strategy. It can be seen that, at earlier rounds, the agent picks a large ratio of students, and only in later rounds, it realizes the critical effect of quarantining workers. Each iteration equals a single run with a population size of 2k, and takes (13.8 ± 5.7) minutes with average computational power.

| Optimization State | Iteration | Students | Workers | Customers | Max Infected People |
|---|---|---|---|---|---|
| A | 0 | 0.4 | 0.3 | 0.7 | 283 |
| B | 4 | 0.6 | 0.3 | 0.5 | 179 |
| C | 6 | 0.6 | 0.2 | 0.6 | 137 |
| D | 16 | 0.5 | 0.6 | 0.3 | 135 |
| E | 21 | 0.6 | 0.4 | 0.4 | 108 |
| F | 31 | 0.7 | 0.1 | 0.6 | 102 |
| G | 55 | 0.7 | 0.5 | 0.2 | 75 |
| H | 73 | 0.6 | 0.6 | 0.2 | 64 |

## Availability and future directions

To ensure the reproducibility of the results presented earlier in the paper, we have published the Pyfectious source code on github.com/amehrjou/Pyfectious. Inside the *cluster* folder, there are instructions (in the form of a Markdown file called REPRODUCTION.md) that briefly explain how our experiments work and how one should be able to reproduce the results and final plots available in the paper. Further, in the *data* folder, more instructions (as README. md files in *data/json* and *data/json/main_experimental_towns* folders) are provided as to how configurations are mapped to the final plots in the paper. We hope that the provided instructions, alongside our comprehensive and ground-up tutorial notebook (colab.research.google. com/drive/14UYB9x0g5s7jyHO4DynqDJMXTaSM11eW), could contribute to making a strong reproducibility case- as well as helping developers design customized and more complex simulations with fewer barriers.

We have introduced Pyfectious, an agent-based individual-level simulation software built upon sophisticated statistical models capable of high-granularity simulation of epidemic disease in a structured population. The modularity and hierarchical structure allow the simulator to quickly adapt to various sizes of the population such as cities, countries, continents, or even worldwide. Thanks to several algorithmic and implementational novelties, such as multi-resolution timelines, Pyfectious can be used on machines with a wide range of computational resources. The control and monitoring modules are designed to facilitate implementing real-world inspired testing and quarantining at various scales from subsets of the population to every individual. These features altogether make Pyfectious a full-fledged environment to search for the most effective policy that controls the spread of the epidemic with minimum economic side effects as we have showcased in this work. Future works can build upon Pyfectious, more sophisticated policy search algorithms that combine an effective representation learning with Monte Carlo Tree Search, such as AlphaZero, to learn the policies that might be impossible for human experts to find due to the immense complexity of the problem.

## Supporting information

**S1 File. Manual simulation.** This example is the best starting point to understand the simulator's software interface and a comprehensive guide to run customized simulations with Pyfectious. A good knowledge of Python is necessary to invoke Pyfectious simulation capacity, especially if one decides to opt for manual simulation.
(PDF)

**S2 File. Configured simulation.** This file is dedicated to explaining the automated configuration process of the simulator. As opposed to manually building the population structure, quarantine strategies, and disease models, the data required by the simulator is obtained from hierarchical JSON configuration files. Using JSON configuration files makes it easier for people with limited knowledge of python, or programming in general, to interact with the software.
(PDF)

**S3 File. Sanity checks.** We demonstrate the procedure of conducting further experiments in order to evaluate the basic functionality and sanity of the simulator. Prior to engaging with this tutorial, one should take a look at S1 File, explaining the manual simulation process.
(PDF)

**S4 File. Additional experiments.** Further experiments have been developed and presented in this file, in order to demonstrate the simulator's ability to cope with a variety of given criteria.

The experiments are fundamentally the same as the main text but vary in terms of parameters like population size or society's structure.
(PDF)

## Author Contributions

**Conceptualization:** Arash Mehrjou.

**Data curation:** Arash Mehrjou, Ashkan Soleymani, Amin Abyaneh.

**Formal analysis:** Arash Mehrjou, Ashkan Soleymani, Amin Abyaneh.

**Funding acquisition:** Bernhard Schölkopf, Stefan Bauer.

**Investigation:** Arash Mehrjou, Ashkan Soleymani, Amin Abyaneh, Samir Bhatt, Stefan Bauer.

**Methodology:** Arash Mehrjou, Ashkan Soleymani.

**Project administration:** Arash Mehrjou, Bernhard Schölkopf, Stefan Bauer.

**Software:** Arash Mehrjou, Ashkan Soleymani, Amin Abyaneh.

**Supervision:** Arash Mehrjou.

**Validation:** Arash Mehrjou, Ashkan Soleymani, Amin Abyaneh.

**Visualization:** Arash Mehrjou, Ashkan Soleymani.

**Writing – original draft:** Arash Mehrjou, Ashkan Soleymani, Amin Abyaneh, Stefan Bauer.

**Writing – review & editing:** Arash Mehrjou, Ashkan Soleymani, Amin Abyaneh, Samir Bhatt, Stefan Bauer.

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
