## [Decision Letter · Decision Letter 0]

9 Dec 2021

Dear 42424242 Bauer,

Thank you very much for submitting your manuscript "Pyfectious: An individual-level simulator to discover optimal containment polices for epidemic diseases" for consideration at PLOS Computational Biology.

As with all papers reviewed by the journal, your manuscript was reviewed by members of the editorial board and by several independent reviewers. In light of the reviews (below this email), we would like to invite the resubmission of a very significantly-revised version that takes into account the reviewers' comments.

Significant changes are required. Please ensure that the manuscript is correctly organised to follow the conventions of Software articles. Also, in your revised manuscript please ensure you highlight the software's contribution to the field and how it is differentiated from the existing tools.

We cannot make any decision about publication until we have seen the revised manuscript and your response to the reviewers' comments. Your revised manuscript is also likely to be sent to reviewers for further evaluation.

Sincerely,

Manja Marz

Software Editor

PLOS Computational Biology

Virginia Pitzer

Deputy Editor-in-Chief

PLOS Computational Biology

Reviewer's Responses to Questions

**Comments to the Authors:**

Reviewer #1: My overall impression of the paper was that it had an abundance of technical details about the software yet could have benefitted from a more focused narrative to support the authors' claims about the work.

The authors made three broad claims:

1. Their approach to building a synthetic population (and other aspects of models) using a hierarchical random variable is more robust against sampling artifacts and gives confidence bounds for decisions based on the simulation results.

2. The simulator is used as the environment of a reinforcement learning problem to find the optimal policies to control the pandemic. The obtained experimental results indicate the simulator's adaptability and capacity in making sound predictions and a successful policy derivation example based on real-world data. As an exemplary application, our results show that the proposed policy discovery method can lead to control measures that produce significantly fewer infected individuals in the population and protect the health system against saturation.

3. Comprehensive documentation and code-snippets of the use cases are provided as Jupyter notebooks to facilitate a quick start in running experiments with an arbitrary setting for epidemic researchers or policymakers.

For claim 1, while I did appreciate the explicit descriptions of the probabilistic aspects of their approach, stochasticity in agent-based models (ABMs) is very common, and there is an abundant literature for sampling approaches that explicitly aim to produce confidence bounds from ABMs, either for propagating input uncertainties with ABC approaches (e.g., Beaumont 2019), Bayesian optimization of stochastic simulators (e.g., Binois et al. 2018), iterated Bayesian filtering (e.g., Pei et al. 2018), and many others (e.g., Thiele et al. 2014). There is also the question about whether their approach improves on other methods for synthetic population construction, which also have stochastic aspects to them, e.g., Gallagher et al. 2018, Cajka et al. 2010, and more recently many others. In addition to not citing relevant comparators, the case was not clearly made that their contribution was novel and useful. In terms of uncertainties and how they can be utilized in supporting decision making, I would point the authors to work in the robust decision making (RDM) literature for examples (e.g., Lempert 2019).

There was very little in the manuscript to support claim 2. Having said that, if the manuscript were to focus on using the software as part of a fully developed reinforcement learning application, applied to a real problem rather than a single proof of concept model that was provided, I think it could be compelling. The simple examples provided throughout Section 4 served as verification of function rather than novel, or compelling use cases, and would likely be more appropriate for a software tutorial.

I agree that the authors did provide sufficient support to claim 3.

I found the literature/model review utilizing Tables 1-4 in the introduction underdeveloped and not particularly helpful in contextualizing the Pyfectious contributions. I'd suggest more clearly describing the criteria used for choosing the cited examples and defining the assessments of each (e.g., how was detail level defined?).

The writing lacked focus and preciseness at times and should be improved for epidemiological content (e.g., see below) and overall clarity (e.g., pg. 4: "Many existing simulators have been developed for a particular population that is determined by their hyper-parameters.").

I've included some additional more minor points below.

Minor points:

pg. 2: "Social distancing policies are often based on expert’s common sense and previous experiences in partially similar conditions [1]. For instance, vaccination [3], travel restrictions [4], school closure [5], and wearing protective instruments such as masks [6] were used during SARS-CoV1 in 2003 whose effect were evaluated through several studies."

 Since vaccination is not a social distancing policy, I suggest to remove it from this sentence about the implementation of social distancing policies.

pg. 7: "edges of a background timer"

 This concept ("edge" in the context of the timer) needs clarification.

pg. 7: "Infection Event" and "Incubation Event"

 For clarity, should these be renamed to something more indicative of the fact that the infection and incubation end when they occur?

pg. 8: "We divide the software system into three separate components that can be considered independently: 1) population generation, 2) disease propagation, and 3) time management."

 At the end of the Introduction section, there was an emphasis on the "two processes in the developed software package," the Population Model and the Propagation Model. It might be helpful to either relate the three components to the two processes or consistently refer to one of the organizing structures throughout.

pg. 12: Regarding the assignment of profession, it seems that order of profession assignment would matter but there was no discussion of how the professions are ordered. Also, the notation at the end of the "Special case of profession assignment" paragraph is unclear.

pg. 12: All the edges appear to be directed in the connectivity graph so in describing the family connections I'd suggest using the same terminology from the Algorithm 2 pseudocode and indicate that the edges are bidirectional. Also, how are the \\alpha_{ij} and \\beta_{ij} meant to be chosen for the Beta distribution, beyond just that the "probability mass is concentrated around 1 for i = j and around smaller values for i \\neq j"?

pg. 14: What are the units for infection rate? Or is it a probability, given it is constrained to be between 0 and 1? I'd suggest clarifying the name and the definition.

pg. 14: Is the transmission potential (\\gamma) modeled as a Gamma or Beta distribution? The text appears to be inconsistent.

pg. 15: "We propose a novel mixture event/clock-driven method to bring together the benefits of both worlds."

 This is not necessarily novel, e.g., see North et al. 2013.

pg. 15: "The queue of events is formed similar to the event-based methods, but the pulses of a background clock determine which events are effective in the outcome."

 This sentence is unclear as to what "which events are effective in the outcome" means.

pg. 15: "Transition event Each transition event is associated with a certain individual, and it is triggered when that individual changes its location from one subcommunity to another. The change of location is implemented by changing the weights of the edges connected to an individual in the connectivity graph."

 Two elements that I think need clarification. 1) Are locations and subcommunities to be thought of as interchangeable concepts? 2) How do edge weights change?

pg. 22: "measuring the required time for the model generation and simulation phases"

 Was the concept of "model generation" introduced previously?

References cited:

Beaumont, M. A. (2019). Approximate Bayesian Computation. Annual Review of Statistics and Its Application, 6(1), 379–403. https://doi.org/10.1146/annurev-statistics-030718-105212

Binois, M., Gramacy, R. B., & Ludkovski, M. (2018). Practical Heteroscedastic Gaussian Process Modeling for Large Simulation Experiments. Journal of Computational and Graphical Statistics, 27(4), 808–821. https://doi.org/10.1080/10618600.2018.1458625

Cajka, J. C., Cooley, P. C., & Wheaton, W. D. (2010). Attribute Assignment to a Synthetic Population in Support of Agent-Based Disease Modeling. Methods Report (RTI Press), 19(1009), 1–14. PubMed. https://doi.org/10.3768/rtipress.2010.mr.0019.1009

Gallagher, S., Richardson, L. F., Ventura, S. L., & Eddy, W. F. (2018). SPEW: Synthetic Populations and Ecosystems of the World. Journal of Computational and Graphical Statistics, 27(4), 773–784. https://doi.org/10.1080/10618600.2018.1442342

Lempert, R. J. (2019). Robust Decision Making (RDM). In V. A. W. J. Marchau, W. E. Walker, P. J. T. M. Bloemen, & S. W. Popper (Eds.), Decision Making under Deep Uncertainty: From Theory to Practice (pp. 23–51). Springer International Publishing. https://doi.org/10.1007/978-3-030-05252-2_2

North, M. J., Collier, N. T., Ozik, J., Tatara, E. R., Macal, C. M., Bragen, M., & Sydelko, P. (2013). Complex Adaptive Systems Modeling with Repast Simphony. Complex Adaptive Systems Modeling, 1(1), 3. https://doi.org/10.1186/2194-3206-1-3

Pei, S., Morone, F., Liljeros, F., Makse, H., & Shaman, J. L. (2018). Inference and control of the nosocomial transmission of methicillin-resistant Staphylococcus aureus. ELife, 7, e40977. https://doi.org/10.7554/eLife.40977

Thiele, J. C., Kurth, W., & Grimm, V. (2014). Facilitating Parameter Estimation and Sensitivity Analysis of Agent-Based Models: A Cookbook Using NetLogo and R. Journal of Artificial Societies and Social Simulation, 17(3), 11. http://jasss.soc.surrey.ac.uk/17/3/11.html

Reviewer #2: The authors define a principled, extensive, open-source environment for running epidemiological simulations, with the ability to define different population structures, conditionals, and event driven milestones.

However, it’s not clear from the repository structure how one would go about reproducing the results found in the paper.

From the Plos guidance: “A prerequisite for publication is that the results described in the paper must be reproducible when peer reviewers or editors choose to run the software on the deposited dataset using the parameters provided.”

Major revisions:

At present the mapping between the results in the paper to a set of simulation configurations is not clear. Making reproducibility a challenge.

e.g. which result does cluster_experiment_0 represent? https://github.com/amehrjou/Pyfectious/tree/master/data/json/cluster_experiment_0

Or inversely, which combination of configuration files represent the experiments highlighted in Figure 13?

Provide such a mapping.

A reconciliation needs to be made between the online documentation and intro guide via python notebook / Google Colaboratory and the supplemental getting started guide in the paper. The link on page 34 in the supplemental section is broken. This is of particular concern as it claims to provide code snippets from the paper.

https://github.com/amehrjou/Pyfectious/tree/master/examples

Given the substantial compute resources necessary to test the results, a reduction in barrier to entry could be made by providing cluster specific launch scripts written for SLURM or other Distributed Resource Management Application API (DRMAA) compliant HPC management system.

Overhead aside, the software does seem like a reasonably well engineered, modular and accessible entry point for those scientists interested in pursuing a counter factual based epidemiological question. It also represents a substantial amount of work. The paper is well written and describes the software well.

Minor suggestions: One limitation appears to be the ability to explicitly track multiple strains of a disease and define a partial immunological relationship between the recovered states of such. Also the out-of-the-box software lacks the ability to define immune response waning as a function of time. Additionally modeling an asymptomatic or pre-symptomatic state would be an interesting modification as it relates to the stated observer functions. These would go along way in terms of addressing current epidemic related concerns.

Reviewer #3: I think Pyfectious represents an interesting addition to the family of agent-based models for infectious disease policy evaluation. There are many that exist already however and this paper does not make a strong case for the novelty of the platform such that i think it merits publication in this journal in its current form.

The EpiSIMS platform (LANL) which you cite and GLEAM from Northeastern Univ and FRED from University of Pittsburgh as well as EpiFast and EpiSimdemics from Virginia Tech all have similar manners of generating synthetic populations and much more elaborate and thus useful representations for disease progression and interventions. Many of these tools have been published on and in widespread use for decades.

The breakthrough your tool represents is its efficiencies allowing for "simulation in the loop" style reinforcement learning as well as the wide array and variety of control measures. While this has also been done by researchers with the above mentioned tools, increasing the efficiency of this process is important and may be a significant. Focusing more on this aspect of the tool and less on the features would be a useful for this manuscript.

The features of this simulator are extensive, however, the main shortfall is that the disease model is very limited (seems to only represent infection and death, whereas hospitalization and symptomatic illness are equally if not more important). Additionally the ability to specify disease parameters for different segments of the population is an essential part of COVID, and does not seem to be a part of this tool (eg mortality rates are much much higher for older individuals, who have different communities etc.). Interventions beyond social distancing are important elements of outbreak control, and the relative timing of diagnoses and treatment are essential. None of features appear to be in this tool, which i think will limit the use cases for this software and this its overall adoption.

Assessing simulation performance in the rather narrow confines of 5K-30K size of population is not enough to determine if the simulation performance is linear, at least 3 orders of magnitude are needed to assess scaling. In my experience detailed agent-based simulations as listed above can simulate towns of this size considerably faster than 100s of minutes for 48 hours, but perhaps i have misunderstood the metrics. I think focusing a little more on the performance would benefit the paper. For example: How do the different "controllers" affect the run time? do certain commands impact the runtime more? What is the full run time for finding an optimal policy? Some of this is addressed in the final section of the paper, but seems more important than the limited time provided.

Several set of experiments are described in the paper, but only one "optimal" policy finding case study presented. I'd be very interested to know what conditions in the population or disease might shift the found optimal solution. An exploration of these kinds of findings (both intuitive and counter-intuitive) would better showcase Pyfectious's benefits.

To summarize, evaluating the performance in finding different optimal social distancing combinations in more detail would be useful to describe Pyfectious's strengths more. Some consideration of pharmaceutical control measures and the difficult tasks of differential timing of case identification and treatment is also important as it is VERY relevant to disease control.

Reviewer #4: The paper describes Pyfectious, a Python package that allows simulation of a disease outbreak at the individual level. It allows creating a probabilistic model of population, simulation of a wide range of policies, and finding optimal control policies using a reinforcement learning framework.

Developing an individual level simulator (e.g., an agent-based model) for epidemiology could take a significant amount of time. The Python packaged described in this paper could be quite useful to public health researchers and allow them to focus their efforts in supplying realistic population parameters (demographics, communities, and connectivity) and evaluating intervention strategies instead of developing code and reduce time. The paper provides reasonably detailed description of the simulation design and examples.

I have following questions:

1. Page 14, the last line of "Transmission potential" mentions Gamma distribution and then uses Beta distribution. Could you please clarify?

2. Page 15, Transition event represents change of location which is implemented by changing weights of edges connected to an individual in the connectivity graph. How are weights determined? Is it based on activity duration (i.e., the amount of time two individuals are at the same location at the same time)?

3. I was wondering if this package could be used to implement contact tracing, e.g., ~50% of the contacts of infected individuals could be traced and put in quarantine for 2 weeks?

4. Page 29, last paragraph, should value of 0.2 for column customers in the last row of table 7 mean that 20% of the customers are restricted and 80% allowed (instead of 20% allowed as mentioned in the text on Page 29)?

5. What are the last two values (True and 1) represent when creating Community_Type_Role 'student_type' on Page 36, section 6.2.2?

Here are some suggestions that could help improve the paper:

1. Second line of the introduction states "While the first category involves medications and vaccination, the second approach, which is the main interest of the current work, concerns interventions on human communities to slow down the spread of the disease." I don't think, this differentiated pharmaceutical and non-pharmaceutical interventions clearly. Both approaches concern interventions on human communities to slow down the spread of a disease. As you mentioned in the first approach involves mediations or vaccines and second approach involves reducing potential for disease transmission by behavioral interventions such as wearing masks, social distancing, or quarantining infected people.

2. The first line second paragraph in the introduction talks about social distancing and next line mentions vaccination as an example. As you mentioned earlier in the introduction, vaccination is not a social distancing intervention.

3. While creating a family pattern, individual demographic distributions are assigned independently. While this is not an unreasonable assumption, in reality, demographics of people in the same family aren't independent of each other and it might be good to acknowledge this. The same argument can be made for activities of family members.

4. Page 3, "Interaction model: The transmission of the disease is through an underlying graph that determines the structure of the population.". Actually, the structure of the population (i.e., demographics, communities, and activities/mobility patterns) determines the underlying graph over which the disease transmission occurs.

5. I think some of the events names may be a bit confusing. For example, the paper mentions "Incubation event occurs when the incubation period is over and indicates a transition from the incubation period to the illness period during which the patient becomes infectious.". I think as this marks start of infectious period it should not be called an incubation event, but may be an infection event. Similarly, as the paper mentions "Infection Event occurs when the infection of an individual ends.". As this marks recovery (or death), may be this should be called Removed event.

6. Section 3.3.4 mentions consistency in results for different clock periods (temporal resolutions). It would be good to show this experimentally.

7. Many of the figures (e.g., Figure 9) show confidence intervals. It would be good to specify exactly what levels of confidence intervals are shown (e.g., 95% confidence intervals).

8. Finding optimal policy could be computationally expensive. It would be good to mention run time for optimal policy estimation for the example in Figure 15 and Table 7.

Minor edits:

1. Page 2, "Population model: We enumerate various aspects of generating the society by creating the individuals."  "Population model: We enumerate various aspects of generating a society by creating a set of individuals."

2. Page 9, section 3.1.1 Personal attributes, "disease, blood pressure, and etc."  "disease, blood pressure, etc."

3. Page 9, section 3.1.2, Should "Its attributes are then are sampled from \\phi_{i} independent of other members." be "Its attributes are then sampled from \\varphi_{i}, independent of other members."?

4. I think, Algorithm 1 should take {\\phi_1, \\phi_2, \\ldots \\phi_n} and {\\pi_1, \\pi_2, \\ldits \\pi_n} as input instead of {\\varphi_1, \\varphi_2, \\ldots, \\varphi_n}. Also, should \\varphi_j in line 5 be \\phi_j?

5. Page 12, should there be infinity symbol in the last line of "Special case of profession assignment" section?

6. Figure 5 caption, "from" is mentioned twice in the last line.

7. Figure 11b, it seems like strategy A and B are swapped between the plot and caption. Strategy B (not A as mentioned in the caption) is the one that enforces quarantine after 20 days and lifts it after 20 days.

8. Figure 12a, it seems like strategies C, D, B, and A (not A, B, C, and D as mentioned in the caption) enforce a quarantine after 40, 60, 100, and 200 days after the outbreak, respectively.

9. Page 34, I could not find https://github.com/amehrjou/Pyfectious/tree/master/example but was able to reach the Google Colaboratory Notebook from the project page on Github. Please update the link in the paper accordingly.

**Have the authors made all data and (if applicable) computational code underlying the findings in their manuscript fully available?**

Reviewer #1: Yes

Reviewer #2: Yes

Reviewer #3: Yes

Reviewer #4: Yes

PLOS authors have the option to publish the peer review history of their article (what does this mean?). If published, this will include your full peer review and any attached files.

Reviewer #1: No

Reviewer #2: No

Reviewer #3: No

Reviewer #4: No
---

## [Decision Letter · Decision Letter 1]

2 Jul 2022

Dear 42424242 Bauer,

Thank you very much for submitting your manuscript "Pyfectious: An individual-level simulator to discover optimal containment polices for epidemic diseases" for consideration at PLOS Computational Biology. As with all papers reviewed by the journal, your manuscript was reviewed by members of the editorial board and by several independent reviewers. The reviewers appreciated the attention to an important topic. Based on the reviews, we are likely to accept this manuscript for publication, providing that you modify the manuscript according to the review recommendations.

Sincerely,

Manja Marz

Software Editor

PLOS Computational Biology

Manja Marz

Software Editor

PLOS Computational Biology

[LINK]

Reviewer's Responses to Questions

**Comments to the Authors:**

Reviewer #2: The authors have made a substantial effort to address reviewer comments. The accessibility of this software make it a valuable entry point for many. The ability to specify the configuration of populations using JSON specifications that capture properties of real world populations is likely to occupy a space for computational epidemiology similar to other powerful yet accessible python libraries.

Reviewer #4: The paper describes Pyfectious, a Python package that allows simulation of a disease outbreak at the individual level. It allows creating a probabilistic model of population, simulation of a wide range of policies, and finding optimal control policies using a reinforcement learning framework. The paper provides detailed description of the system and has addressed previous reviewer comments.

Major comments:

1. The first paragraph on Page 25 (The case of Multi-Resolution Simulation) states following: “Results in low-resolution (T=320 mins) is asymptotically consistent with very-high resolution regimes (T=40 mins).” It is not clear what they mean by “asymptotically consistent” here. There is a large difference in the number of infections over time (as well as peak and total number of infections) between T=40 mins and T=320 mins regimes, which could lead to different conclusions when the clock period is changed. My understanding is that one of the claims of the paper is that Pyfectious can provide similar results by changing clock period (or adjusting computational requirements) but some of the important results/numbers do not seem to hold when clock period is adjusted.

Minor edits:

1. The last line in the first paragraph of the introduction seems a bit off. May be something like following could be more suitable: “Considering the vaccine policy as behavioral intervention allows investigation of situations where there is a shortage of vaccine supply or the population is not fully vaccinated along with other behavioral interventions such as social distancing.”

2. The second paragraph on Page 3 states “We tried to compare with a diverse set of highly-respected simulators in terms of their attitude and simulation methods in different pieces of literature, ..”. It is not clear what attitude refers to here.

3. Page 3, last paragraph, a reference is missing (??).

4. Page 4, interaction model: “The transmission of the disease is through an underlying graph that is determined the structure of the population.”  “The transmission of the disease is through an underlying graph that is determined by the structure of the population.”

5. Page 5, first paragraph: “Pyfectious allows a much richer space of possible policies, even including probabilistic and conditional control which also is not suffering from interpretation ambiguities.”  “Pyfectious allows a much richer space of possible policies, including probabilistic and conditional control which do not suffer from interpretation ambiguities.”

6. Page 5, Policy discovery, last sentence: “Similar way, control theory…”  “Similarly, control theory...”

7. Page 9: “Infection Event the event is queued once”  “Infection Event is queued once”

8. Page 9, last paragraph and Page 14, first paragraph in "Propagation of the infection" section refer to Section 2.2.1 and line 10. Line 10 from where (which algorithm)?

**Have the authors made all data and (if applicable) computational code underlying the findings in their manuscript fully available?**

Reviewer #2: Yes

Reviewer #4: Yes

PLOS authors have the option to publish the peer review history of their article (what does this mean?). If published, this will include your full peer review and any attached files.

Reviewer #2: No

Reviewer #4: No

Figure Files:

Data Requirements:

Reproducibility:

References:

---

## [Decision Letter · Decision Letter 2]

8 Dec 2022

Dear Mr. Mehrjou,

We are pleased to inform you that your manuscript 'Pyfectious: An individual-level simulator to discover optimal containment policies for epidemic diseases' has been provisionally accepted for publication in PLOS Computational Biology.

Best regards,

Manja Marz

Software Editor

PLOS Computational Biology

Jason Papin

Editor-in-Chief

PLOS Computational Biology

Feilim Mac Gabhann

Editor-in-Chief

PLOS Computational Biology

Reviewer's Responses to Questions

**Comments to the Authors:**

Reviewer #4: The authors have redone experiments to explain multi-resolution aspect of the tool well and have addressed all reviewer comments. I think this paper/package will be a valuable addition to the community.

**Have the authors made all data and (if applicable) computational code underlying the findings in their manuscript fully available?**

Reviewer #4: Yes

PLOS authors have the option to publish the peer review history of their article (what does this mean?). If published, this will include your full peer review and any attached files.

Reviewer #4: No

---

## [Editor Report · Acceptance letter]

13 Jan 2023

PCOMPBIOL-D-21-00718R2 

Pyfectious: An individual-level simulator to discover optimal containment policies for epidemic diseases

Dear Dr Mehrjou,

I am pleased to inform you that your manuscript has been formally accepted for publication in PLOS Computational Biology. Your manuscript is now with our production department and you will be notified of the publication date in due course.

With kind regards,

Anita Estes
